# Robust Satisficing Gaussian Process Bandits Under Adversarial Attacks

**Artun Saday** [*]
Bilkent University, Ankara, Türkiye
artun.saday@bilkent.edu.tr

**Yaşar Cahit Yıldırım** [*]
Bilkent University, Ankara, Türkiye
cahit.yildirim@bilkent.edu.tr

**Cem Tekin**
Bilkent University, Ankara, Türkiye
cemtekin@ee.bilkent.edu.tr

## Abstract

We address the problem of Gaussian Process (GP) optimization in the presence of unknown and potentially varying adversarial perturbations. Unlike traditional robust optimization approaches that focus on maximizing performance under worst-case scenarios, we consider a robust satisficing objective, where the goal is to consistently achieve a predefined performance threshold $\tau$, even under adversarial conditions. We propose two novel algorithms based on distinct formulations of robust satisficing, and show that they are instances of a general robust satisficing framework. Further, each algorithm offers different guarantees depending on the nature of the adversary. Specifically, we derive two regret bounds: one that is sublinear over time, assuming certain conditions on the adversary and the satisficing threshold $\tau$, and another that scales with the perturbation magnitude but requires no assumptions on the adversary. Through extensive experiments, we demonstrate that our approach outperforms the established robust optimization methods in achieving the satisficing objective, particularly when the ambiguity set of the robust optimization framework is inaccurately specified.

## 1 Introduction

Bayesian optimization (BO) is a framework particularly suited for sequentially optimizing difficult-to-evaluate black-box functions $f$, using a probabilistic model such as a Gaussian process (GP) [1]. BO uses a probabilistic surrogate model (e.g., GP) with an acquisition function to decide where to sample next. Designing acquisition functions tailored to specific problems is a significant research area in the BO literature. The performance of a BO algorithm is often evaluated by its *regret*, which quantifies the difference between the objective function values at the chosen points and the optimal point, or another predefined success criterion.

BO has proven effective in applications such as hyperparameter tuning [2], engineering design [3], experimental design [4], and clinical modeling [5]. In these settings, robustness to distribution shifts and environmental variability is essential. For instance, machine learning models must generalize beyond training data, while engineering designs must transfer from simulation to real-world deployment. In high-stakes domains like autonomous driving or patient treatment, ignoring uncertainty can lead to severe failures. Robust optimization (RO) addresses this by seeking solutions that perform well under worst-case scenarios defined by an ambiguity set [6]. However, the ambiguity set must be carefully defined: if it is too narrow, it may exclude relevant uncertainties; if too broad, it can lead to conservative solutions that underperform under typical conditions [7, 8].

---

[*]Equal contribution.

39th Conference on Neural Information Processing Systems (NeurIPS 2025).

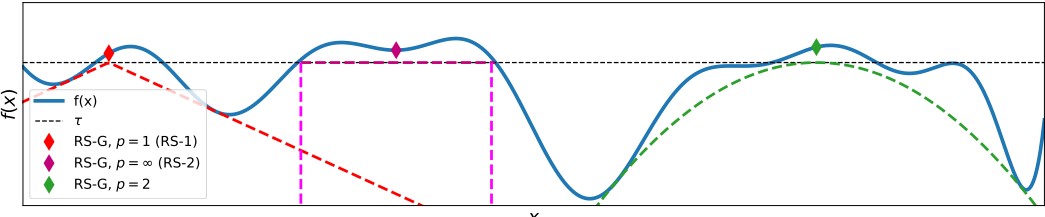

Figure 1: Illustration of how different RS formulations trade off reward guarantees with respect to perturbation magnitudes during action selection. Decision of each formulation can be interpreted as selecting the action (diamonds) by enforcing the widest fragility-cone (dashed lines) to limit the performance degradation beyond $\tau$ due to the perturbation. The fragility-cone is the reward guarantee of the solution $x^{\text{RS-G}}$: $\tau - [\kappa_{\tau,p} d(x^{\text{RS-G}}, x^{\text{RS-G}} + \delta)]^p$, defined in (10). It characterizes the shape of the constraint on performance degradation. For more details, see Section 4 and Appendix B.

Satisficing, a term coined by Herbert Simon [9], introduces a pragmatic approach to decision-making where the goal is to achieve an outcome $x$ that is *good enough* ($f(x) \geq \tau$) rather than optimal. This concept is especially relevant in environments where the optimal solution is either unattainable or impractical due to computational or informational constraints. Satisficing has been studied in the context of multi-armed bandits (MAB) by [10, 11, 12, 13], while [14] investigates the problem in a non-Bayesian setting using the RKHS framework. Building on this concept, robust satisficing (RS) extends the principle of satisficing by aiming to optimize confidence in achieving a sufficiently good outcome, thus providing a robust framework for decision-making under uncertainty. Unlike simple satisficing, which may serve as a stopping heuristic, RS focuses on ensuring reliable performance across a variety of scenarios. Decision theorists such as [15] and [16] argue that RS offers a superior alternative to RO, as it emphasizes acceptable performance across a broad range of scenarios rather than focusing solely on worst-case outcomes. [17] proposes an optimization formulation for RS which inspires our approach. [18] builds on this by combining RO and RS in a unified framework that accounts for both outcome and estimation uncertainties. In our work we extend the RS literature by proposing a novel formulation, RS-G, which generalizes [17] by allowing nonlinear decay of the reward guarantee via a tunable parameter.

One form of uncertainty that RO and RS seek to address are adversarial perturbations. In many learning systems, small, targeted changes to the input can induce large deviations in output, undermining reliability. While this phenomenon is well studied in neural networks [19, 20], similar vulnerabilities exist in continuous control and optimization settings, where the selected input may be corrupted by noise or external interference. Addressing such perturbations requires robust strategies that extend beyond worst-case formulations, particularly when the perturbation model or budget is only partially known. This motivates our development of an RS approach for adversarially perturbed GP optimization.

**Related Works** Research on robustness, particularly adversarial robustness, in GP optimization is extensive. [21] tackles BO with outliers by combining robust GP regression with outlier diagnostics. [22], closely related to our work, introduces the setting of adversarial robustness in GP bandits. They propose an RO based algorithm that assumes a known perturbation budget, while we adopt an RS approach without this assumption. [23] addresses a similar problem where the learner selects $\tilde{x}_t$ but observes outputs at a different point $x_t \sim P_{\tilde{x}_t}$, using a GP model for probability distributions. In the noisy input setting, [24] proposes an entropy search algorithm. Adversarial corruption of observations, rather than inputs, is considered by [25, 26]. Contextual GP optimization considers uncertainty in the context distribution. [27] address this using RO with a known ambiguity set, while [28] extend the approach by adopting RS without assuming knowledge of the ambiguity set. A complementary direction is explored by [29, 30, 31], focusing on the BO of risk measures while [32] jointly optimizes the mean and variance of $f(x, \omega)$ using a GP.

**Contributions** We propose new RS-based acqusition functions for the first time for a variant of the GP optimization problem called *Adversarially Robust GP Optimization* introduced in [22], where an adversary perturbs the selected action. Based on principles of robust satisficing, we introduce two novel algorithms for finding a robust action while achieving small lenient and robust satisficing

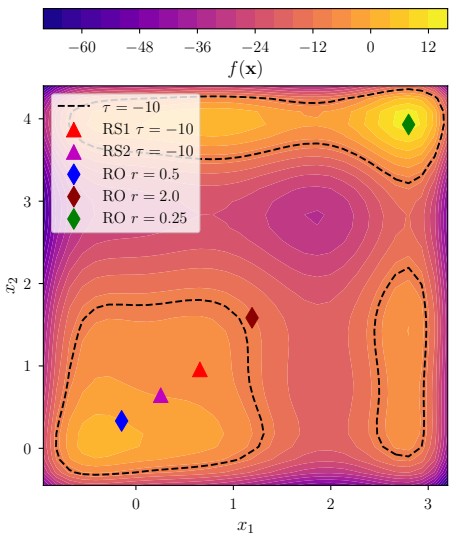

| Objective | Formulation |
|-----------|-------------|
| Standard | $\arg\max\limits_{x} f(x)$ |
| RO | $\arg\max\limits_{x} \min\limits_{\delta \in \Delta_\epsilon(x)} f(x + \delta)$ |
| RS-1 | $\arg\min\limits_{x} k$ s.t. 
 $f(x + \delta) \geq \tau - kd(x,\, x + \delta),$ 
 $\forall \delta \in \Delta_\infty(x),\ \ k \geq 0$ |
| RS-2 | $\arg\max\limits_{x} \epsilon$ s.t. 
 $f(x + \delta) \geq \tau,\ \ \forall \delta \in \Delta_\epsilon(x),$ 
 $\epsilon \geq 0$ |
| RS-G | $\arg\min\limits_{x} k$ s.t. 
 $f(x + \delta) \geq \tau - [kd(x,\, x + \delta)]^p,$ 
 $\forall \delta \in \Delta_\infty(x),\ \ k \geq 0$ |

Figure 2: **(Left)** Illustration of RO and RS solutions. Highlighted contours represent $f(x) = \tau$, i.e., the satisficing threshold. **(Right)** Formulations of different optimization objectives.

regret, based on two distinct RS formulations (RS-1 and RS-2 in Figure 2). We show that these formulations are edge cases of a general robust satisficing framework (RS-G in Figure 2 (right)) with smoothness parameter $p \geq 1$. We demonstrate how a spectrum of robust solutions can be obtained by varying $p$ (see Figure 1). We provide theoretical bounds on the regrets of our algorithms under certain assumptions. Finally, we present experiments, including in-silico experiments on an FDA-approved simulator for diabetes management with realistic adversarial perturbations, to demonstrate the strengths of our algorithms addressing the shortcomings of other approaches known in the literature.

## 2 Problem Formulation

Let $f$ be an unknown function defined on the domain $\mathcal{X} \subset \mathbb{R}^m$, endowed with a pseudometric $d(\cdot, \cdot) : \mathcal{X} \times \mathcal{X} \to \mathbb{R}$. Further assume that $f$ belongs to a Reproducing Kernel Hilbert Space (RKHS) $\mathcal{H}$, with the reproducing kernel $k(\cdot, \cdot) : \mathcal{X} \times \mathcal{X} \to \mathbb{R}$, and has bounded Hilbert norm $\|f\|_{\mathcal{H}} \leq B$, for some $B > 0$. This assumption ensures that $f$ satisfies smoothness conditions such as Lipschitz continuity with respect to the kernel metric[2] [33] and allows the construction of rigorous confidence bounds using GPs [34]. At each round $t \in [T]$, where $T$ represents the time horizon, the learner picks a point $\tilde{x}_t \in \mathcal{X}$, in response to which the adversary selects a perturbation $\delta_t$. The learner then observes the perturbed point $x_t = \tilde{x}_t + \delta_t$ along with the noisy sample at that point $y_t = f(x_t) + \eta_t$, where $\eta_t$ are independent $R$-subGaussian random variables. If the adversary is allowed to play any perturbation $\delta_t$, no matter what action $\tilde{x}_t$ the learner chooses, the adversary can always choose a perturbation $\delta_t$ such that $\tilde{x}_t + \delta_t = \arg\min_{x \in \mathcal{X}} f(x)$, and the optimization problem would be hopeless. Hence at each round $t$, we assume an *unknown perturbation budget* $\epsilon_t$, that bounds set of perturbations the adversary can play. We define for each point $x \in \mathcal{X}$, the *ambiguity ball*

$$\Delta_\epsilon(x) := \{x' - x : x' \in \mathcal{X} \text{ and } d(x, x') \leq \epsilon\} . \tag{1}$$

The ambiguity ball (1) defines the set of all permissible perturbations of the adversary, when the perturbation budget is $\epsilon$. Hence in response to action $\tilde{x}_t$, the adversary plays the perturbation $\delta_t \in \Delta_{\epsilon_t}(\tilde{x}_t)$.

*Remark* 2.1. In the presented setting, it is not possible to learn the function optimum in general. As an example consider the two arm MAB problem with $\mathcal{X} = \{x_1, x_2\}$ with $f(x_1) = 0$ and $f(x_2) = 1$.

---

[2]For the kernel metric, we have $|f(x) - f(x')| = |\langle f, k(\cdot, x) - k(\cdot, x')\rangle| \leq \|f\|_{\mathcal{H}} d_k(x, x')$. Note that $f$ is not necessarily Lipschitz continuous with respect to an arbitrary metric on $\mathcal{X}$. Hence in our analysis, we assume the pseudometric $d(\cdot, \cdot)$ is the kernel metric. Additionally, when the feature map $k(\cdot, x)$ is injective, this pseudometric is also a metric.

Then at each round $t$, a sufficiently strong adversary can always choose a perturbation $\delta_t$ such that $x_t = x_1$, hence the learner can never observe the optimal action.

## 2.1 Robust Satisficing

The key difference between RS and RO lies in their objectives. While RO seeks to optimize for worst-case outcomes, RS prioritizes achieving satisfactory performance across various scenarios. RO relies heavily on defining an ambiguity set, typically as a ball with radius $r$, centered around a reference value of the contingent variable. The choice of $r$ greatly impacts performance. Overestimating $r$ leads to conservative solutions that overemphasize unlikely worst-case scenarios, while underestimating $r$ risks overly optimistic solutions. Figure 2 (left) demonstrates the effect of different choices of $r$.

In our context, the contingent variables are the adversarial perturbations. In other settings, such as data-driven optimization with distributional shifts, contingent variables might represent different uncertainties, such as deviations from an empirical distribution [35]. Unlike RO, which requires an ambiguity set, RS relies only on the satisficing threshold $\tau$, representing the desired outcome. This makes $\tau$ more interpretable and adaptable to real-world problems, where it can be chosen by domain experts. We now present two formulations based on the RS approach, assuming the function $f$ is known.

**RS-1** We adapt the novel formulation by [17] to our adversarial setting. Let $(\cdot)^+ = \max\{\cdot, 0\}$. The objective of this formulation is to find $x^{\text{RS-1}} \in \mathcal{X}$ that solves the optimization problem $\kappa_\tau := \min_{x \in \mathcal{X}} \kappa_\tau(x)$ where $\kappa_\tau(x)$ is the *fragility* of action $x \in \mathcal{X}$ defined as

$$\kappa_\tau(x) := \min k \quad \text{s.t.} \quad f(x+\delta) \geq \tau - kd(x, x+\delta), \quad \forall \delta \in \Delta_\infty(x), \quad k \geq 0 \,. \tag{2}$$

This is equivalent to solving

$$\kappa_\tau(x) := \begin{cases} \left( \max_{\delta \in \Delta_\infty(x) \backslash 0} \frac{\tau - f(x+\delta)}{d(x, x+\delta)} \right)^+ & \text{if } f(x) \geq \tau \\ +\infty & \text{if } f(x) < \tau. \end{cases} \tag{3}$$

When (2) is feasible, the RS-1 action is defined as $x^{\text{RS-1}} = \arg\min_{x \in \mathcal{X}} \kappa_\tau(x)$. The fragility of an action can be seen as the minimum rate of suboptimality the action can achieve w.r.t. the threshold $\tau$, per unit perturbation, for all possible perturbations. Crucially, while the RO formulation only considers perturbations possible under a given perturbation budget $\epsilon$, the RS-1 formulation (2) considers all possible perturbations. This way RS gives safety guarantees even when the perturbation budget of the adversary is unknown. Figure 2 (left) highlights the RS-1 action $x^{\text{RS-1}}$ together with optimization objectives.

**RS-2** Inspired by the literature on robust satisficing, we present another formulation called RS-2, similar to the one in [36]. Formally, the objective of RS-2 is to find $x^{\text{RS-2}} \in \mathcal{X}$ that solves the following optimization problem $\epsilon_\tau := \max_{x \in \mathcal{X}} \epsilon_\tau(x)$ where $\epsilon_\tau(x)$ is the *critical radius* of action $x \in \mathcal{X}$ defined as

$$\epsilon_\tau(x) := \max_{x' \in \mathcal{X}} d(x, x') \quad \text{s.t.} \quad f(x+\delta) \geq \tau, \quad \forall \delta \in \Delta_{d(x,x')}(x) \,. \tag{4}$$

The robust satisficing action is defined as the one with the maximum critical radius, $x^{\text{RS-2}} = \arg\max_{x \in \mathcal{X}} \epsilon_\tau(x)$. If $f(x) = \tau$ and $x$ is a local maximum, then $\epsilon_\tau(x) = 0$ because $\delta = 0$ is the only perturbation satisfying the inequality condition. If $f(x) < \tau$, then $\epsilon_\tau(x) = -\infty$, as no $\epsilon$ satisfies the condition, and by convention, we take the supremum of the empty set as the critical radius, which is $-\infty$. Unlike RO, RS-2 does not assume knowledge of the adversary's budget $\epsilon_t$ but instead searches for the action that achieves the threshold $\tau$ under the widest possible range of perturbations. This target-driven approach aligns RS-2 with the philosophy of robust satisficing. Figure 2 (left) highlights $x^{\text{RS-2}}$ alongside other optimization objectives. If $\tau > f(\hat{x})$, where $\hat{x} := \arg\max_{x \in \mathcal{X}} f(x)$ is the optimal action, no action can meet the threshold, even without perturbations, hence we introduce the following assumption moving on.

**Assumption 2.2.** $\tau \leq \max_{x \in \mathcal{X}} f(x)$.

Assumption 2.2 ensures that the optimization problems (2) for RS-1 and (4) for RS-2 are feasible, thereby guaranteeing that a robust satisficing action is well-defined in both cases. However, since $f$ is

unknown, it may not be immediately clear to the learner whether a specific threshold $\tau$ satisfies this assumption. If the learner is flexible with the satisficing threshold, Assumption 2.2 can be relaxed by dynamically selecting $\tau$ at each round to be less than $\mathrm{lcb}_t(\hat{x}'_t)$, where $\hat{x}'_t := \arg\max_{x \in \mathcal{X}} \mathrm{lcb}_t(x)$. The function $\mathrm{lcb}_t$ represents the lower confidence bound of the function, whose proper definition is given in Section 3. Additionally, in many real-world tasks, $\tau$ can be chosen to satisfy Assumption 2.2 using domain-specific information. For example, in hyperparameter tuning where the objective $f(x)$ represents model accuracy, $f(\hat{x})$ is upper bounded by 1 (the maximum possible accuracy) and can be lower bounded by the accuracy of the best-known model (e.g., 0.97).

*Remark* 2.3. Our formulation treats the perturbation as an arbitrary adversarial quantity rather than as a random variable. Therefore, direct comparison with probabilistic robustness measures such as Value-at-Risk (VaR), Conditional Value-at-Risk (CVaR), or distributionally robust optimization (DRO) is not immediate, since these formulations assume access to a known or estimable perturbation distribution. Incorporating such measures would require modifying the problem setting to include a random covariate in the objective, $f(x, \xi)$, and redefining the satisficing condition with respect to the distribution of $\xi$. While one could, for example, define a variant of robust satisficing conditioned on a reference estimate of this distribution, such an approach restricts the power of the adversary and leads to a less general setting than the one considered here. We emphasize these conceptual differences to clarify the distinction between adversarial and probabilistic robustness.

## 2.2 GP Regression

It is well established in the literature that a function $f \in \mathcal{H}$ from an RKHS with reproducing kernel $k(\cdot, \cdot)$ with bounded norm $\|f\|_{\mathcal{H}} \leq B$ can be approximated using GPs. Given a dataset of observation pairs $\mathcal{D}_t = \{x_i, y_i\}_{i=1}^t$, using a Gaussian likelihood with variance $\lambda > 0$ and a prior distribution $\mathrm{GP}(0, k(x, x'))$, the posterior mean and covariance of a Gaussian process can be calculated as

$$\mu_t(x) = \boldsymbol{k}_t(x)^\mathsf{T}(\boldsymbol{K}_t + \lambda \boldsymbol{I}_t)^{-1}\boldsymbol{y}_t, \quad k_t(x, x') = k(x, x') - \boldsymbol{k}_t(x)^\mathsf{T}(\boldsymbol{K}_t + \lambda \boldsymbol{I}_t)^{-1}\boldsymbol{k}_t(x'), \quad (5)$$

with $\sigma_t^2(x) = k_t(x, x)$, where column vector $\boldsymbol{k}_t(x) = [k(x_i, x)]_{i=1}^t$, $\boldsymbol{K}_t$ is the $t \times t$ kernel matrix with elements $(\boldsymbol{K}_t)_{ij} = k(x_i, x_j)$. These posterior mean and variance functions enable the construction of confidence intervals for the function $f(x)$, which are integral to the optimization strategies we employ. Specifically, in our adversarial setting, these confidence intervals are used to bound the fragility $\kappa_\tau(x)$ and the critical radius $\epsilon_\tau(x)$ of each action, as discussed in the next section.

## 2.3 Regret Measures

In BO it is common to measure the success of an algorithm using a *regret measure*, which usually measures the discrepancy between the reward of the action played against the reward of the optimal action. We use *lenient regret* and *robust satisficing regret* defined respectively as:

$$R_T^l := \sum_{t=1}^T (\tau - f(x_t))^+, \qquad R_T^{rs} := \sum_{t=1}^T (\tau - \kappa_\tau \epsilon_t - f(x_t))^+. \tag{6}$$

Lenient regret, introduced by [11], is a natural metric for satisficing objectives. It captures the cumulative loss with respect to the satisficing threshold. Sublinear lenient regret is unattainable under unconstrained adversarial attacks, as a sufficiently large perturbation budget allows the adversary to force any action below $\tau$. Thus, upper bounds on lenient regret must depend on perturbation magnitude or impose constraints on the budget $\epsilon_t$ and threshold $\tau$. Robust satisficing regret [28], instead evaluates loss relative to the RS-1 benchmark, comparing the true reward of $x_t$ against the guaranteed reward of $x^{\mathrm{RS-1}}$ under perturbations of at most $\epsilon_t$. The RS-1 action ensures $f(x^{\mathrm{RS-1}}) \geq \tau - \kappa_\tau \epsilon_t$ under such a budget.

## 3  Algorithms and Theoretical Results

**Confidence Intervals**  Our algorithms utilize the GP-based upper and lower confidence bounds defined as $\mathrm{ucb}_t(x) = \mu_{t-1}(x) + \beta_t \sigma_{t-1}(x)$ and $\mathrm{lcb}_t(x) = \mu_{t-1}(x) - \beta_t \sigma_{t-1}(x)$, respectively, where $\beta_t$ is a $t$-dependent exploration parameter. For theoretical guarantees, $\beta_t$ is set as in Lemma 3.1, which is based on [37] but argued to be less conservative and more practical [38].

**Lemma 3.1.** *[38, Theorem 1] Let $\zeta \in (0,1)$, $\overline{\lambda} := \max\{1, \lambda\}$, and*

$$\beta_t(\zeta) := B + \frac{R}{\sqrt{\lambda}}\sqrt{\log(\det(\frac{\overline{\lambda}}{\lambda}\boldsymbol{K}_{t-1} + \overline{\lambda}\boldsymbol{I}_{t-1})) + 2\log\left(\frac{1}{\zeta}\right)}.$$

*Then, the following holds WPAL $1 - \zeta$:* $\text{lcb}_t(x) \leq f(x) \leq \text{ucb}_t(x)$, $\forall x \in \mathcal{X}$, $\forall t \geq 1$ .

We label the event in Lemma 3.1 as $\mathcal{E}$ and refer to it as the *good event*. Our analyses will make use of the *maximum information gain* defined as $\gamma_t := \max_{A \subset \mathcal{X}:|A|=t} \frac{1}{2}\log(\det(\boldsymbol{I}_t + \lambda^{-1}\boldsymbol{K}_A))$, where $\boldsymbol{K}_A$ is the kernel matrix given by the sampling set $A$. Following the work of [34], the use of $\gamma_t$ is common in the GP literature. Since $f$ is an unknown black-box function, $\kappa_\tau$ for RS-1 and $\epsilon_\tau$ for RS-2 cannot be calculated directly. The proposed algorithms leverage upper and lower confidence bounds to estimate these values while balancing the exploration-exploitation trade-off.

---

**Algorithm 1** AdveRS-1

---

1: **Input:** Kernel function $k$, $\mathcal{X}$, $\tau$, $R$, $B$, confidence parameter $\zeta$, time horizon $T$
2: **Initialize:** Set $\mathcal{D}_0 = \emptyset$ (empty dataset), $\mu_0(x) = 0$, $\sigma_0(x) = 1$ for all $x$
3: **for** $t = 1$ to $T$ **do**
4:     Compute $\text{ucb}_t(x) = \mu_{t-1}(x) + \beta_t\sigma_{t-1}(x)$, $\forall x \in \mathcal{X}$
5:     Compute $\overline{\kappa}_{\tau,t}(x)$ as in (7), $\forall x \in \mathcal{X}$
6:     Select point $\tilde{x}_t = \arg\min_{x \in \mathcal{X}} \overline{\kappa}_{\tau,t}(x)$
7:     Adversary selects perturbation $\delta_t \in \Delta_{\epsilon_t}(\tilde{x}_t)$
8:     Sample $x_t = \tilde{x}_t + \delta_t$, observe $y_t = f(x_t) + \eta_t$
9:     Update dataset $\mathcal{D}_t = \mathcal{D}_{t-1} \cup \{(x_t, y_t)\}$
10:     Update GP posterior as in (5)
11: **end for**

---

**Algorithm 2** AdveRS-2

---

5:     Compute $\overline{\epsilon}_{\tau,t}(x)$ as in (8), $\forall x \in \mathcal{X}$
6:     Select point $\tilde{x}_t = \arg\max_{x \in \mathcal{X}} \overline{\epsilon}_{\tau,t}(x)$

---

**Algorithm for RS-1.** To perform BO with the objective of RS-1, we propose *Adversarially Robust Satisficing-1* (AdveRS-1) algorithm, whose pseudo code is given in Algorithm 1. At the beginning of each round $t$, AdveRS-1 computes the upper confidence bound $\text{ucb}_t(x)$ of each action $x \in \mathcal{X}$. Using the $\text{ucb}_t$ in (2), it computes the *optimistic fragility* defined as:

$$\overline{\kappa}_{\tau,t}(x) := \left(\max_{\delta \in \Delta_\infty(x)\backslash 0} \frac{\tau - \text{ucb}_t(x + \delta)}{d(x, x + \delta)}\right)^+ \tag{7}$$

if $\text{ucb}_t(x) \geq \tau$, and otherwise $\overline{\kappa}_{\tau,t}(x) := \infty$. Further, we define the minimizer $\overline{\kappa}_{\tau,t} := \min_{x \in \mathcal{X}} \overline{\kappa}_{\tau,t}(x)$. The optimistic fragility, $\overline{\kappa}_{\tau,t}(x)$, provides an optimistic estimate of the true fragility $\kappa_\tau(x)$ for each action. AdveRS-1 selects the action with the smallest optimistic fragility, $\tilde{x}_t = \arg\min_{x \in \mathcal{X}} \overline{\kappa}_{\tau,t}(x)$, to encourage exploration. Additionally, we define the *pessimistic fragility* $\underline{\kappa}_{\tau,t}(x)$ which uses lcb instead of the ucb in equation (7), and $\underline{\kappa}_{\tau,t} := \min_{x \in \mathcal{X}} \underline{\kappa}_{\tau,t}(x)$. The following lemma relates the estimated and true fragilities of an action.

**Lemma 3.2.** *Given $\mathcal{E}$ holds, the inequality, $\overline{\kappa}_{\tau,t}(x) \leq \kappa_\tau(x) \leq \underline{\kappa}_{\tau,t}(x)$ is true for all $x \in \mathcal{X}$, $t \geq 1$.*

Lemma 3.2 is a result of the monotonicity of fragility result in [17], with the proof provided in the Appendix. The following Corollary which illustrate how $\underline{\kappa}_{\tau,t}(x)$ and $\overline{\kappa}_{\tau,t}(x)$ contribute to quantifying the robustness of an action $x$ at round $t$, follows from Lemma 3.2 and the RS-1 objective (2).

**Corollary 3.3.** *Under Assumption 2.2, with probability at least $1 - \zeta$, $\forall t \in [T]$ it holds that:* $f(\tilde{x}_t + \delta) \geq \tau - \underline{\kappa}_{\tau,t} \cdot d(\tilde{x}_t, \tilde{x}_t + \delta)$, $\forall \delta \in \Delta_\infty(\tilde{x}_t)$; *conversely,* $\exists \delta' \in \Delta_\infty(\tilde{x}_t)$ *s.t.* $f(\tilde{x}_t + \delta') \leq \tau - \overline{\kappa}_{\tau,t} \cdot d(\tilde{x}_t, \tilde{x}_t + \delta')$.

**Algorithm for RS-2** For the RS-2 objective, we follow a similar path and propose the *Adversarially Robust Satisficing-2* (AdveRS-2) algorithm which is given in Algorithm 2. The algorithm follows

a similar structure to Algorithm 1 with only the lines that differ being shown, along with their corresponding line numbers. AdveRS-2, again, makes use of the confidence bounds to create an optimistic estimate of the critical radius $\epsilon_\tau(x)$ of each action. Specifically, we define the optimistic critical radius as:

$$\bar{\epsilon}_{\tau,t}(x) := \max_{x' \in \mathcal{X}} d(x, x') \text{ s.t. } \text{ucb}(x + \delta) \geq \tau, \quad \forall \delta \in \Delta_{d(x,x')}(x) , \tag{8}$$

with $\bar{\epsilon}_{\tau,t} := \max_{x \in \mathcal{X}} \bar{\epsilon}_{\tau,t}(x)$. AdveRS-2 uses the optimistic critical radius for exploration, sampling at each round $t$ the action $\tilde{x}_t = \arg\max_{x \in \mathcal{X}} \bar{\epsilon}_{\tau,t}(x)$, which has the largest optimistic critical radius. Further we define the *pessimistic critical radius* $\underline{\epsilon}_\tau(x)$, which uses the lcb instead of the ucb in equation (8). The following Lemma bounds the true critical radius $\epsilon_\tau(x)$ of each action.

**Lemma 3.4.** *Given $\mathcal{E}$ holds, the inequality, $\underline{\epsilon}_{\tau,t}(x) \leq \epsilon_\tau(x) \leq \bar{\epsilon}_{\tau,t}(x)$ is true for all $x \in \mathcal{X}$, $t \geq 1$.*

**Corollary 3.5.** *Under Assumption 2.2, with probability at least $1 - \zeta$, $\forall t \in [T]$, it holds that $f(\tilde{x}_t + \delta) \geq \tau$, $\forall \delta \in \Delta_{\underline{\epsilon}_{\tau,t}}(\tilde{x}_t)$ and that when $\Delta_{\epsilon_\tau}(\tilde{x}_t)$ is a strict subset of $\Delta_{\bar{\epsilon}_{\tau,t}}(\tilde{x}_t)$, $\exists \delta \in \Delta_{\bar{\epsilon}_{\tau,t}}(\tilde{x}_t)$ s.t. $f(\tilde{x}_t + \delta) \leq \tau$.*

Corollary 3.5 gives information about the achievability of satisficing threshold $\tau$. In practice, a learner might use this information to tune the $\tau$ either to a more ambitious or to a more conservative one.

**Regret analysis of AdveRS-1**  Regret analysis of Algorithm 1 is done with the distance function defined as the kernel metric, $d(x, x') = \sqrt{k(x, x) - 2k(x, x') + k(x', x')}$, and we make use of the Lipschitz continuity of $f$.

**Theorem 3.6.** *Let $\zeta \in (0, 1)$, and let $f \in \mathcal{H}$ with $\|f\|_{\mathcal{H}} \leq B$. Let $k(\cdot, \cdot)$ be the reproducing kernel of $\mathcal{H}$, and let $\eta_t$ be conditionally $R$-subgaussian. Under Assumption 2.2, when the distance function $d(\cdot, \cdot)$ is the kernel metric and $\beta_t$ as defined in Lemma 3.1, the lenient regret and robust satisficing regret of AdveRS-1 are bounded above by:*

$$R_T^l \leq 4\beta_T \sqrt{T\gamma_T} + B \sum_{t=1}^{T} \epsilon_t , \qquad R_T^{rs} \leq 4\beta_T \sqrt{T\gamma_T} . \tag{9}$$

**Proposition 3.7.** *In the context of Theorem 3.6, the linear term $B \sum_{t=1}^{T} \epsilon_t$ in the lenient regret bound is unavoidable. Specifically, there exists an instance of the problem setup where this term manifests as a lower bound on the lenient regret for any algorithm.*

**Regret analysis of AdveRS-2**  The RS-1 formulation provides reward guarantees under any perturbation $\delta \in \Delta_\infty(x^{\text{RS-1}})$, ensuring that $f(x^{\text{RS-1}}) \geq \tau - \kappa_\tau \cdot d(x^{\text{RS-1}}, x^{\text{RS-1}} + \delta)$. In contrast, the RS-2 formulation does not provide reward guarantees for every possible perturbation but offers a stronger guarantee, $f(x^{\text{RS-2}}) \geq \tau$, for a restricted set of perturbations $\delta \in \Delta_{\epsilon_\tau}(x^{\text{RS-2}})$ where $\epsilon_\tau$ is defined in (4). This distinction is reflected in the regret analysis of AdveRS-2. To bound the regret of AdveRS-2, we introduce the following assumption.

**Assumption 3.8.** Assume that the perturbation budget of the adversary $\epsilon_t \leq \epsilon_\tau$ for all $t \geq 1$.

*Remark* 3.9. Assumption 3.8, which is stronger than Assumption 2.2, ensures that at each round $t \leq T$, there exists an action $x \in \mathcal{X}$ meeting the threshold $\tau$ against any adversary-selected perturbation $\delta_t \in \Delta_{\epsilon_t}(x)$. Without Assumption 3.8, achieving sublinear lenient regret is impossible, as there will always be a perturbation $\delta \in \Delta_{\epsilon_t}(x)$ causing $f(x + \delta) \leq \tau$ for any $x$.

**Theorem 3.10.** *Under the assumptions of Theorem 3.6 and Assumption 3.8, both the lenient regret and the robust satisficing regret of AdveRS-2 are bounded above by: $R_T^l, R_T^{rs} \leq 4\beta_T \sqrt{T\gamma_T}$ .*

While the lenient regret bound of AdveRS-2 requires further assumptions on the perturbation budget of the adversary, it is also stronger than the regret bound of AdveRS-1 and matches the standard regret bound $\tilde{\mathcal{O}}(\gamma_T \sqrt{T})$ of GP-UCB [37]. When the kernel $k(\cdot, \cdot)$ is the RBF kernel or the Mátern-$\nu$ kernel, the regret bound reduces to $\tilde{\mathcal{O}}(\sqrt{T})$ and $\tilde{\mathcal{O}}(T^{\frac{2\nu+3m}{4\nu+2m}})$, respectively, where $m$ is the dimension of the input space [39].

## 4  Unifying the RS Formulations

The reward guarantee of the RS-1 solution in (2) falls off linearly with the magnitude of the perturbation. On the other hand the reward guarantee of the RS-2 solution stays constant up to a perturbation

magnitude of $\epsilon_\tau$, than disappears. In practice one might want to be more flexible with the reward guarantee of their solution. For example one might desire the solution obtained to give a guarantee that stays close to $\tau$ for small perturbations, while not giving too much importance to larger perturbations as they may be deemed unlikely, depending on the structure of the problem. Motivated by this, we propose the following robust satisficing formulation RS-General (RS-G). For $p \geq 1$ define the $p$-fragility of an action as,

$$\kappa_{\tau,p}(x) := \min k \quad \text{s.t.} \quad f(x + \delta) \geq \tau - [kd(x, x + \delta)]^p, \quad \forall \delta \in \Delta_\infty(x), \quad k \geq 0 . \quad (10)$$

The RS-G action is $x^{\text{RS-G}} := \arg\min_{x \in \mathcal{X}} \kappa_{\tau,p}(x)$.

**Proposition 4.1.** *Let $x^{RS\text{-}G}$ denote the solution obtained from the RS-G formulation when applied with a power parameter $p$. It holds that $\lim_{p \to \infty} x^{RS\text{-}G} = x^{RS\text{-}2}$.*

The parameter $p$ controls the trade-off in the RS-G formulation. Specifically, at $p = 1$, the reward guarantee decreases linearly with perturbation magnitude. As $p$ increases, it enhances reward guarantees for small perturbations but diminishes for large perturbations. As $p$ approaches infinity, the formulation converges to RS-2, offering robust guarantees for perturbations within $\Delta_{\epsilon_\tau}(x^{\text{RS-2}})$ but none for those outside it. A more detailed comparison of the RS-G formulation with RS-1 and RS-2 is available in Appendix B. Algorithm 1 can be readily adapted to the RS-G framework by replacing optimistic fragilities with optimistic $p$-fragilities $\overline{\kappa}_{\tau,p,t}(x)$, which are (10) calculated on $\text{ucb}_t(x)$, instead of $f(x)$. We call this general algorithm AdveRS-G. Similarly, we can generalize our RS regret to the RS-G formulation $R_T^{rs\text{-}g} := \sum_{t=1}^{T} (\tau - [\kappa_{\tau,p}\epsilon_t]^p - f(x_t))^+$. This measures, at each time step the discrepancy, between the played action, and the reward guarantee of the $x^{\text{RS-G}}$ solution, for a specific $p$. Finally, we bound the general RS regret in the next corollary.

**Corollary 4.2.** *Under the assumptions of Theorem 3.6, AdveRS-G satisfies $R_T^{rs\text{-}g} \leq 4\beta_T \sqrt{T\gamma_T}$.*

## 5 Experiments

In each experiment, we compare the performance of AdveRS-1 and AdveRS-2 against the RO baseline [22], which is run with an ambiguity ball of radius $r$ that is equal to, smaller than, and greater than the true perturbation budget $\epsilon_t$ at each round and use the Euclidean distance as the perturbation metric. We use the following attack schemes in our experiment:

- **Random Attack:** $\delta_t \sim \text{Uniform}(\Delta_{\epsilon_t}(\tilde{x}_t))$
- **LCB Attack:** $\delta_t = \arg\min_{\delta \in \Delta_{\epsilon_t}(\tilde{x}_t)} \text{lcb}_t(\tilde{x}_t + \delta)$
- **Worst Case Attack:** $\delta_t = \arg\min_{\delta \in \Delta_{\epsilon_t}(\tilde{x}_t)} f(\tilde{x}_t + \delta)$

The results of all experiments are averaged over 100 runs, with error bars representing std/2. Additional experiments can be found in the Appendix E. Our code is available at: `https://github.com/Bilkent-CYBORG/AdveRS`.

**Synthetic experiment** In the first experiment, we use the proof-of-concept function in Figure 2, a modified version of the synthetic function from [22]. We set the threshold $\tau = -10$ and conduct two experiments: (a) $\epsilon_t = 0.5$ (Assumption 3.8 holds) and (b) $\epsilon_t = 1.5$ (Assumption 3.8 *fails* to hold). For the RO representative, we run STABLEOPT [22] with radius parameters $r = \epsilon_t$, $r = 4\epsilon_t$, and $r = 0.5\epsilon_t$. The observation noise follows $\eta_t \sim \mathcal{N}(0, 1)$, and the GP kernel is a polynomial kernel trained on 500 samples from the function. Figure 3a shows that STABLEOPT performs poorly when the adversary's perturbation budget is misestimated, leading to linear regret, while AdveRS-2 consistently meets the threshold $\tau$, achieving sublinear lenient regret. AdveRS-1, though not always reaching $\tau$, still outperforms STABLEOPT when $r$ is misspecified. As depicted in Figure 3b, when Assumption 3.8 fails, all algorithms experience linear regret, consistent with Remark 3.9. This figure also illustrates that AdveRS algorithms maintain robustness towards the goal $\tau$ even when $\tau$ is unattainable. Additionally, RS-G with $p = 2$ exhibits the smallest regret, suggesting that tuning the $p$ parameter can be beneficial for certain problems. Figure 4 our algorithms consistently beat the RO baseline in RS regret, even when STABLEOPT is run with the perfect knowledge of the adversarial budget.

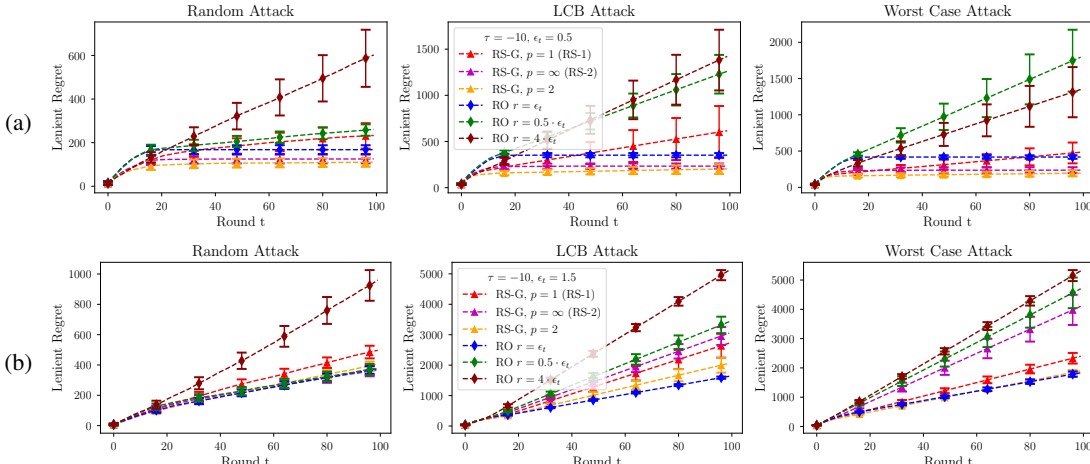

(a)

(b)

Figure 3: Lenient regret results for synthetic experiment shown in two scenarios: (a) satisfying and (b) failing to satisfy Assumption 3.8.

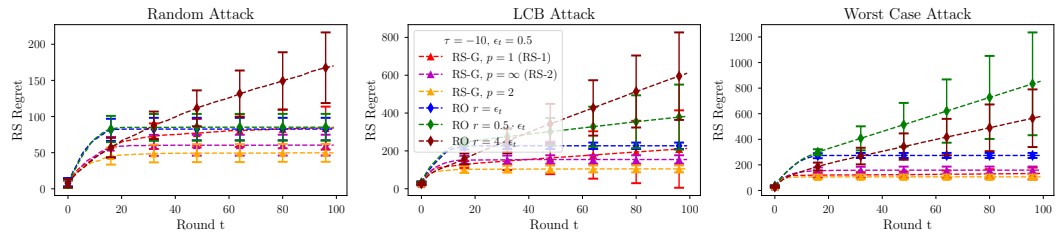

Figure 4: Robust satisficing regret results for synthetic experiment.

**Insulin dosage** In adversarial contextual GP optimization, the learner selects an action $x_t \in \mathcal{X}$ after observing a reference contextual variable $c_t^{\text{ref}} \in \mathcal{C}$, which is then perturbed by an adversary. Then, the learner observes the perturbed context $c_t = c_t^{\text{ref}} + \delta_t$ along with the noisy observation $f(x_t, c_t) + \eta_t$. We apply this to an insulin dosage selection problem for Type 1 Diabetes Mellitus (T1DM) patients using the UVA/PADOVA T1DM simulator [40]. T1DM patients require bolus insulin administrations usually taken before a meal [41]. Contextual information about the carbohydrate intake is known to improve postprandial blood glucose (PBG) prediction [42].

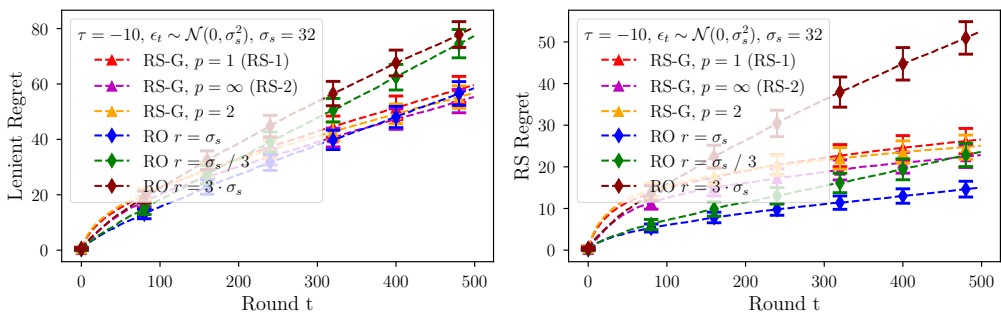

Figure 5: Lenient and RS regret results of insulin dosage experiment.

However, the carbohydrate announcements prior to meal consumption can deviate from the actual carbohydrate intake, hence robustness to contextual shifts are important in diabetes management. Maintaining PBG around a $\tau$ neighbourhood of a target level $K$ as $[K - \tau, K + \tau]$ mg/dL represents a satisficing objective defined as $-|f(x) - K| \geq \tau$. The action space is units of insulin in the range $[0, 15]$, and the context is carbohydrate intake, perturbed by $\delta_t \sim \mathcal{N}(0, \sigma_s^2)$ with $\sigma_s = 32$,

representing roughly 2 slices of bread. As shown in Figure 10, although no algorithm achieves sublinear regret, both AdveRS-1 and AdveRS-2 outperform STABLEOPT when the radius $r$ is suboptimal. While STABLEOPT performs best when $r = \sigma_s$, selecting this parameter can be difficult. In contrast, the parameter $\tau$ can be more easily chosen by a clinician, aligning with common clinical practice [41].

**Robustness to worst case attack**   We consider the robustness of the solutions from Figure 2 (left), under worst case adversarial perturbations of increasing magnitude. Specifically we inspect $\min_{\delta \in \Delta_\epsilon(x^*)} f(x^* + \delta)$ where $x^*$ corresponds to the points selected by each objective. To evaluate the robustness of a solution w.r.t. achieving $\tau$, we calculate the following:

$$\text{Area}(\epsilon) = \int_0^\epsilon \max \left( 0, \tau - \min_{\delta \in \Delta_{\epsilon'}(x^*)} f(x^* + \delta) \right) d\epsilon'.$$

The lower this value is, the better the robustness of the method to a range of different perturbations up to $\epsilon$. A clear discrepancy can be seen between our methods and the RO baseline, with both RS-1 and RS-2 actions demonstrating greater robustness. Further, while RS-2 is better in achieving $\tau$, RS-1 is more robust to discrepancies when $\tau$ is not attainable.

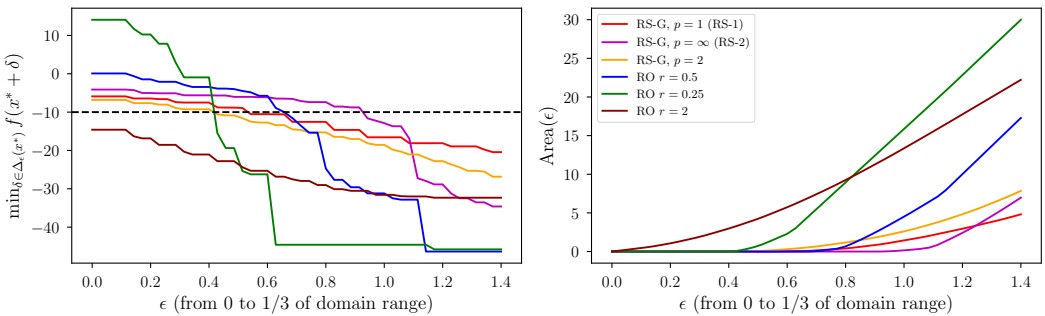

Figure 6: **(Left)** Rewards for the points selected by each algorithm, under worst case attack with perturbation budget $\epsilon$. **(Right)** Area under the curves as a function of perturbation magnitudes.

## 6   Conclusion

In this paper, we addressed GP optimization under adversarial perturbations, where both the perturbation budget and strategy are unknown and variable. Traditional RO approaches rely on defining an ambiguity set, which is challenging with uncertain adversarial behavior. We introduced a robust satisficing framework with two formulations, RS-1 and RS-2, aimed at achieving a performance threshold $\tau$ under these conditions. We also showed these two formulations can be united under a more general RS formulation. The RS-1-based algorithm offers a lenient regret bound that scales with perturbation magnitude and a sublinear robust satisficing regret bound. The RS-2-based algorithm provides sublinear lenient and robust satisficing regret guarantees, assuming $\tau$ is achievable. Our experiments show that both algorithms outperform robust optimization methods, particularly when the ambiguity set in RO is misestimated. Despite its promise, our framework has limitations. Selecting $\tau$ can be challenging without domain expertise, and the regret bounds for AdveRS-2 assume that $\tau$ is achievable. Future work could focus on adaptive $\tau$ selection and applying the RS formulations to broader settings, such as distribution shifts and supervised learning. Additionally our framework focuses on the case with known adversarial perturbations. A promising future direction is to develop an RS approach in a setting where the perturbations are not observed. Another future direction would be to learn the parameter $p$ in the RS-G formulation during algorithm run time.

**Acknowledgements:** This work was supported by the Scientific and Technological Research Council of Türkiye (TÜBİTAK) under Grant 124E065; TÜBİITAK 2024 Incentive Award; by the Turkish Academy of Sciences Distinguished Young Scientist Award Program (TÜBA-GEBİP-2023). Y. Cahit Yıldırım was supported by Turk Telekom as part of 5G and Beyond Joint Graduate Support Programme coordinated by Information and Communication Technologies Authority.

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

# A  Table of Notations

| Notation | Description |
|---|---|
| $\mathcal{X}$ | Action set |
| $\tau$ | satisficing level (threshold) |
| $d(\cdot,\cdot) : \mathcal{X} \times \mathcal{X} \to \mathbb{R}$ | Distance metric on $\mathcal{X}$ |
| $\Delta_\epsilon(x)$ | $\{x' - x : \mathbf{x}' \in \mathcal{X} \text{ and } d(x,x') \leq \epsilon\}$, set of perturbations in $\epsilon$-ball |
| $\hat{x}$ | $\arg\max_{x\in\mathcal{X}} f(x)$ |
| $\tilde{x}_t$ | Action selected by each algorithm |
| $x_t$ | Point sampled after perturbation, $\tilde{x}_t + \delta_t$ |
| $\delta_t$ | Perturbation of the adversary at round $t$ |
| $\epsilon_t$ | True perturbation budget of the adversary at time $t$ |
| $\kappa_\tau(x)$ | $\begin{cases} \left(\max_{\delta\in\Delta_\infty(x)\setminus 0} \frac{\tau - f(x+\delta)}{d(x,x+\delta)}\right)^+ & \text{if } f(x) \geq \tau \\ +\infty & \text{if } f < \tau \end{cases}$ |
| $\kappa_\tau$ | $\min_{x\in\mathcal{X}} \kappa_\tau(x)$ |
| $x^{\text{RS-1}}$ | $\arg\min_{x\in\mathcal{X}} \kappa_{\tau,t}(x)$ |
| $\overline{\kappa}_{\tau,t}(x)$ | $\begin{cases} \left(\max_{\delta\in\Delta_\infty(x)\setminus 0} \frac{\tau - \text{ucb}_t(x+\delta)}{d(x,x+\delta)}\right)^+ & \text{if } \text{ucb}_t(x) \geq \tau \\ +\infty & \text{if } \text{ucb}_t(x) < \tau \end{cases}$ |
| $\overline{\kappa}_{\tau,t}$ | $\min_{x\in\mathcal{X}} \overline{\kappa}_{\tau,t}(x)$ |
| $\underline{\kappa}_{\tau,t}(x)$ | $\begin{cases} \left(\max_{\delta\in\Delta_\infty(x)\setminus 0} \frac{\tau - \text{lcb}_t(x+\delta)}{d(x,x+\delta)}\right)^+ & \text{if } \text{lcb}_t(x) \geq \tau \\ +\infty & \text{if } \text{lcb}_t(x) < \tau \end{cases}$ |
| $\underline{\kappa}_{\tau,t}$ | $\min_{x\in\mathcal{X}} \underline{\kappa}_{\tau,t}(x)$ |
| $\epsilon_\tau(x)$ | $\max_{x'\in\mathcal{X}} d(x,x')$ s.t. $f(x+\delta) \geq \tau,\ \forall \delta \in \Delta_\epsilon(x)$ |
| $\epsilon_\tau$ | $\max_{x\in\mathcal{X}} \epsilon_\tau(x)$ |
| $x^{\text{RS-2}}$ | $\arg\max_{x\in\mathcal{X}} \epsilon_\tau(x)$ |
| $\overline{\epsilon}_{\tau,t}(x)$ | $\max_{x'\in\mathcal{X}} d(x,x')$ s.t. $\text{ucb}_t(x+\delta) \geq \tau,\ \forall \delta \in \Delta_\epsilon(x)$ |
| $\overline{\epsilon}_{\tau,t}$ | $\max_{x\in\mathcal{X}} \overline{\epsilon}_{\tau,t}(x)$ |
| $\underline{\epsilon}_{\tau,t}(x)$ | $\max_{x'\in\mathcal{X}} d(x,x')$ s.t. $\text{lcb}_t(x+\delta) \geq \tau,\ \forall \delta \in \Delta_\epsilon(x)$ |
| $\underline{\epsilon}_{\tau,t}$ | $\max_{x\in\mathcal{X}} \underline{\epsilon}_{\tau,t}(x)$ |

Table 1: Table of notations

# B  Unified Formulation RS-G

The $p$-fragility $\kappa_{\tau,p}(x)$ of action $x \in \mathcal{X}$ is

$$\kappa_{\tau,p}(x) := \min k \text{ s.t. } f(x+\delta) \geq \tau - [kd(x,x+\delta)]^p,\ \forall \delta \in \Delta_\infty(x) .$$

This means the RS-G action $x_p^{\text{RS-G}}$ gives the reward guarantee of $f(x+\delta) \geq \tau - [\kappa_{\tau,p} d(x_p^{\text{RS-G}}, x_p^{\text{RS-G}} + \delta)]^p$. When $p = 1$, this guaratee is a linear cone facing downward from $\tau$, at the choosen point. Since the $p$-fragility is is the minimum $k$ that satisfies this inequality, this means the guarantee of the $x_p^{\text{RS-G}}$ action can be represented by the widest cone drawn from $\tau$. When $p = 2$, instead of a linear cone, the guarantee takes quadratic form and so on. We generalize the notion of a cone for all $p$ and define the *fragility-cone* of an action $x' \in \mathcal{X}$ as the following function: $\text{Cone}_{x'}^p(x) := \tau - [\kappa_{\tau,p} d(x', x)]^p$. Then we can say that the solution $x^{\text{RS-G}}$ which minimizes $\kappa_{\tau,p}$, is the solution that maximizes the *wideness* of the fragility-cone, as $\kappa_{\tau,p}$ controls how wide the cone is. Figure 1 gives a visual representation of RS-1, RS-2 and RS-G with $p = 2$ solutions, and their respective fragility-cones. Which formulation should be used depends on the structure of the problem and the goal of the optimizer. If, for example large perturbations are considered unlikely, RS-G with a larger $p$ might be better suited.

## C  Proofs

### C.1  Proof of Lemma 3.2

Assume the good event $\mathcal{E}$ holds. Consider the first inequality $\overline{\kappa}_{\tau,t}(x) \leq \kappa_\tau(x)$. When $\overline{\kappa}_{\tau,t}(x) = \infty$, then $\kappa_\tau(x) = \infty$ as well. When $\overline{\kappa}_{\tau,t}(x) < \infty$ and $\kappa_\tau(x) = \infty$, the inequality holds. Let $\overline{\delta}_{x,t} := \arg\max_{\delta \in \Delta_\infty(x) \backslash 0} \frac{\tau - \mathrm{ucb}_t(x+\delta)}{d(x,x+\delta)}$, $\underline{\delta}_{x,t} := \arg\max_{\delta \in \Delta_\infty(x) \backslash 0} \frac{\tau - \mathrm{lcb}_t(x+\delta)}{d(x,x+\delta)}$ and $\delta_{x,t} := \arg\max_{\delta \in \Delta_\infty(x) \backslash 0} \frac{\tau - f(x+\delta)}{d(x,x+\delta)}$. When $\overline{\kappa}_{\tau,t}(x), \kappa_\tau(x) < \infty$ we have

$$
\begin{aligned}
\overline{\kappa}_{\tau,t}(x) - \kappa_\tau(x) &= \frac{\tau - \mathrm{ucb}_t(x + \overline{\delta}_{x,t})}{d(x, x + \overline{\delta}_{x,t})} - \frac{\tau - f(x + \delta_{x,t})}{d(x, x + \delta_{x,t})} \\
&\leq \frac{\tau - \mathrm{ucb}_t(x + \overline{\delta}_{x,t})}{d(x, x + \overline{\delta}_{x,t})} - \frac{\tau - f(x + \overline{\delta}_{x,t})}{d(x, x + \overline{\delta}_{x,t})} \\
&= \frac{f(x + \overline{\delta}_{x,t}) - \mathrm{ucb}_t(x + \overline{\delta}_{x,t})}{d(x, x + \overline{\delta}_{x,t})} \\
&\leq 0 \,.
\end{aligned}
$$

Similarly for $\kappa_\tau(x) \leq \underline{\kappa}_{\tau,t}(x)$ we have

$$
\begin{aligned}
\kappa_{\tau,t}(x) - \underline{\kappa}_\tau(x) &= \frac{\tau - f(x + \delta_{x,t})}{d(x, x + \delta_{x,t})} - \frac{\tau - \mathrm{lcb}_t(x + \underline{\delta}_{x,t})}{d(x, x + \underline{\delta}_{x,t})} \\
&\leq \frac{\tau - f(x + \delta_{x,t})}{d(x, x + \delta_{x,t})} - \frac{\tau - \mathrm{lcb}_t(x + \delta_{x,t})}{d(x, x + \delta_{x,t})} \\
&= \frac{\mathrm{lcb}(x + \delta_{x,t}) - f(x + \delta_{x,t})}{d(x, x + \delta_{x,t})} \\
&\leq 0 \,.
\end{aligned}
$$

$\square$

### C.2  Proof of Lemma 3.4

Assume the good event $\mathcal{E}$ holds. Consider the inequality $\epsilon_\tau(x) \leq \overline{\epsilon}_{\tau,t}(x)$. Assume that this inequality is false for some $x$, then $\exists \delta$ s.t. $\delta \in \Delta_{\epsilon_\tau(x)}(x)$ and $\delta \notin \Delta_{\overline{\epsilon}_{\tau,t}(x)}(x)$ since $\Delta_{\overline{\epsilon}_{\tau,t}(x)}(x) \subset \Delta_{\epsilon_\tau(x)}(x)$, and $f(x + \delta) \geq \tau > \mathrm{ucb}_t(x + \delta)$, which is a contradiction. Similarly assume $\underline{\epsilon}_{\tau,t}(x) \leq \epsilon_\tau(x)$ is false for some $x$. Then $\exists \delta$ s.t. $\delta \in \Delta_{\underline{\epsilon}_{\tau,t}(x)}(x)$ and $\delta \notin \Delta_{\epsilon_\tau(x)}(x)$ since $\Delta_{\epsilon_\tau(x)}(x) \subset \Delta_{\underline{\epsilon}_{\tau,t}(x)}(x)$, and $\mathrm{lcb}_t(x + \delta) \geq \tau > f(x + \delta)$, which is a contradiction. Therefore Lemma 3.4 is true.

### C.3  Proof of Theorem 3.6

**Lenient Regret Bound of AdveRS-1**  The regret bound of AdveRS-1 follows a similar structure to the bound in [28], but instead of the MMD metric, we utilize the Lipschitz continuity of $f$. Define the instantaneous regret at round $t$ as $r_t^l := \tau - f(\tilde{x}_t + \delta_t)$. The cumulative regret is $R_T^l = \sum_{t=1}^{T}(r_t^l)^+$. Assume that the good event $\mathcal{E}$ holds. Then the instantaneous regret can be bounded as

$$
r_t^l = \tau - f(\tilde{x}_t + \delta_t) \leq \tau - \mathrm{ucb}_t(\tilde{x}_t + \delta_t) + 2\beta_t \sigma_t(\tilde{x}_t + \delta_t) \,. \tag{11}
$$

If the adversary does not perturb the action, i.e., $\delta_t = 0$, then we have

$$
r_t^l \leq \tau - \mathrm{ucb}_t(\tilde{x}_t) + 2\beta_t \sigma_t(\tilde{x}_t)
$$

by Assumption 2.2 and the selection rule of the algorithm we have $\mathrm{ucb}_t(\tilde{x}_t) \geq \tau$, hence

$$
r_t^l \leq 2\beta_t \sigma_t(\tilde{x}_t) = 2\beta_t \sigma_t(x_t) \,.
$$

If $\delta_t \neq 0$, continuing from (11),

$$
r_t^l \leq \tau - \mathrm{ucb}_t(\tilde{x}_t + \delta_t) + 2\beta_t \sigma_t(\tilde{x}_t + \delta_t) \tag{12}
$$

$$
\leq d(\tilde{x}_t, \tilde{x}_t + \delta_t) \frac{\tau - \mathrm{ucb}_t(\tilde{x}_t + \delta_t)}{d(\tilde{x}_t, \tilde{x}_t + \delta_t)} + 2\beta_t \sigma_t(\tilde{x}_t + \delta_t) \tag{13}
$$

$$
\leq d(\tilde{x}_t, \tilde{x}_t + \delta_t) \overline{\kappa}_{\tau,t}(\tilde{x}_t) + 2\beta_t \sigma_t(\tilde{x}_t + \delta_t) \tag{14}
$$

$$
\leq d(\tilde{x}_t, \tilde{x}_t + \delta_t) \overline{\kappa}_{\tau,t}(\hat{x}) + 2\beta_t \sigma_t(\tilde{x}_t + \delta_t) \tag{15}
$$

Define $\bar{\delta}_{x,t} := \arg\max_{\delta \in \Delta_\infty(x)\setminus 0} \frac{\tau - \mathrm{ucb}_t(x+\delta)}{d(x,x+\delta)}$. Then we can write above as,

$$= d(\tilde{x}_t, \tilde{x}_t + \delta_t)\frac{\tau - \mathrm{ucb}_t(\hat{x} + \bar{\delta}_{\hat{x},t})}{d(\hat{x}, \hat{x} + \bar{\delta}_{\hat{x},t})} + 2\beta_t\sigma_t(\tilde{x}_t + \delta_t) \tag{16}$$

$$\leq d(\tilde{x}_t, \tilde{x}_t + \delta_t)\frac{f(\hat{x}) - f(\hat{x} + \bar{\delta}_{\hat{x},t})}{d(\hat{x}, \hat{x} + \bar{\delta}_{\hat{x},t})} + 2\beta_t\sigma_t(\tilde{x}_t + \delta_t) \tag{17}$$

$$\leq d(\tilde{x}_t, \tilde{x}_t + \delta_t)\frac{Bd(\hat{x}, \hat{x} + \bar{\delta}_{\hat{x},t})}{d(\hat{x}, \hat{x} + \bar{\delta}_{\hat{x},t})} + 2\beta_t\sigma_t(\tilde{x}_t + \delta_t) \tag{18}$$

$$\leq 2\beta_t\sigma_t(\tilde{x}_t + \delta_t) + B\epsilon_t = 2\beta_t\sigma_t(x_t) + B\epsilon_t . \tag{19}$$

(14) comes from the definition (7) and (15) follows from $\tilde{x}_t$ being the minimizer of $\overline{\kappa}_{\tau,t}(x)$. (17) follows from Assumption 2.2 and the confidence bounds. Finally (18) follows from the Lipschitz continuity of $f$ with respect to the kernel metric $d$, and (19) from $d(\tilde{x}_t, \tilde{x}_t + \delta_t) \leq \epsilon_t$. From this point on, we bound the lenient regret by following standard steps for bounding regret of GP bandits. First note that since (19) is nonnegative, it is also a bound for $(r_t^l)^+$. By plugging this upper bound to (6), and using monotonicity of $\beta_t$, we obtain

$$R_T^l \leq 2\beta_T \sum_{t=1}^{T} \sigma_t(x_t) + B\sum_{t=1}^{T} \epsilon_t \tag{20}$$

$$\leq 2\beta_T \sqrt{T\sum_{t=1}^{T} \sigma_t(x_t)^2} + B\sum_{t=1}^{T} \epsilon_t \tag{21}$$

where (21) uses the Cauchy-Schwartz inequality. Next, to relate the sum of variances that appears in (21) to the maximum information gain, we use $\alpha \leq 2\log(1 + \alpha)$ for all $\alpha \in [0, 1]$ to obtain

$$\sum_{t=1}^{T} \sigma_t(x_t)^2 \leq \sum_{t=1}^{T} 2\log(1 + \sigma_t(x_t)^2)$$
$$\leq 4\gamma_T , \tag{22}$$

where (22) follows from [37, Lemma 3]. Putting (22) in the regret definition we get the final bound

$$R_T^l \leq 4\beta_T\sqrt{T\gamma_T} + B\sum_{t=1}^{T} \epsilon_t .$$

$\square$

**Robust Satisficing Regret Bound of AdveRS-1** Define the instantaneous regret at round $t$ as $r_t^{rs} := \tau - \kappa_\tau\epsilon_t - f(\tilde{x}_t + \delta_t)$. The cumulative regret is $R_T^{rs} = \sum_{t=1}^{T}(r_t^{rs})^+$. Assume that the good event $\mathcal{E}$ holds. Then the instantaneous regret can be bounded as

$$r_\tau^{rs} \leq \tau - \kappa_\tau\epsilon_t - \mathrm{ucb}_t(\tilde{x}_t + \delta_t) + 2\beta_t\sigma_t(\tilde{x}_t + \delta_t) \tag{23}$$
$$\leq \tau - \kappa_\tau\epsilon_t - (\tau - \overline{\kappa}_{\tau,t}(\tilde{x}_t)d(\tilde{x}_t, \tilde{x}_t + \delta_t)) + 2\beta_t\sigma_t(\tilde{x}_t + \delta_t) \tag{24}$$
$$\leq \tau - \kappa_\tau\epsilon_t - (\tau - \overline{\kappa}_{\tau,t}(\tilde{x}_t)\epsilon_t) + 2\beta_t\sigma_t(\tilde{x}_t + \delta_t) \tag{25}$$
$$= \epsilon_t(\overline{\kappa}_{\tau,t} - \kappa_\tau) + 2\beta_t\sigma_t(\tilde{x}_t + \delta_t) \tag{26}$$
$$\leq 2\beta_t\sigma_t(\tilde{x}_t + \delta_t) = 2\beta_t\sigma_t(x_t) . \tag{27}$$

(24) follows from the guarantee of the RS formulation $\mathrm{ucb}_t(\tilde{x}_t + \delta) \geq \tau - \overline{\kappa}_{\tau,t}d(\tilde{x}_t, \tilde{x}_t + \delta)$ for any $\delta \in \Delta_\infty(\tilde{x}_t)$. (27) follows from $\overline{\kappa}_{\tau,t} \leq \kappa_{\tau,t}$ since $\mathrm{ucb}_t(x) \geq f(x)$ for all $x$. Then we bound the cumulative robust satisficing regret, following the same standard steps as in the proof of the cumulative lenient regret bound, to obtain an upper bound of:

$$R_T^{rs} \leq 4\beta_T\sqrt{T\gamma_T} .$$

$\square$

## C.4 Proof of Theorem 3.10

Under the good event $\mathcal{E}$, Assumption 2.2 and Assumption 3.8, we can bound the lenient regret of AdveRS-2 as:

$$r_\tau^l := \tau - f(\tilde{x}_t + \delta_t) \tag{28}$$

$$\leq \min_{\delta \in \Delta_{\bar{\epsilon}_{\tau,t}}(\tilde{x}_t)} \text{ucb}_t(\tilde{x}_t + \delta) - f(\tilde{x}_t + \delta_t) \tag{29}$$

$$\leq \min_{\delta \in \Delta_{\epsilon_\tau}(\tilde{x}_t)} \text{ucb}_t(\tilde{x}_t + \delta) - f(\tilde{x}_t + \delta_t) \tag{30}$$

$$\leq \min_{\delta \in \Delta_{\epsilon_t}(\tilde{x}_t)} \text{ucb}_t(\tilde{x}_t + \delta) - f(\tilde{x}_t + \delta_t) \tag{31}$$

$$\leq \text{ucb}_t(\tilde{x}_t + \delta_t) - f(\tilde{x}_t + \delta_t) \tag{32}$$

$$\leq 2\beta_t \sigma_t(x_t) . \tag{33}$$

Note that from Assumption 2.2, under $\mathcal{E}$ we have $\text{ucb}_t(\hat{x}) \geq \tau$ which means $\bar{\epsilon}_{\tau,t}(\hat{x}) \geq 0$. Then by definition, for any $\delta \in \Delta_{\bar{\epsilon}_t}(\tilde{x}_t)$, $\text{ucb}_t(\tilde{x}_t + \delta) \geq \tau$, which gives (29). (30) is true because $\epsilon_\tau \leq \bar{\epsilon}_{\tau,t}(x^{\text{RS-2}})$ by Lemma 3.4 and $\bar{\epsilon}_{\tau,t}(x^{\text{RS-2}}) \leq \bar{\epsilon}_{\tau,t}$ by the equation (8). Hence $\Delta_{\epsilon_\tau}(\tilde{x}_t) \subseteq \Delta_{\bar{\epsilon}_{\tau,t}}(\tilde{x}_t)$. Similarly, (31) follows from Assumption 3.8. For (32), observe that $\delta_t \in \Delta_{\epsilon_t}(\tilde{x}_t)$, no matter the attacking strategy of the adversary. Finally (33) holds under the good event $\mathcal{E}$. The rest of the proof is the same as the previous ones. Noting that $R^{rs} \leq R^l$ by definition, and hence shares the same upper bound completes the proof of Theorem 3.10. □

## C.5 Proof of Corollary 3.3

By the definition of $\underline{\kappa}_{\tau,t}(x)$, we have $\text{lcb}_t(\tilde{x}_t + \delta) \geq \tau - \underline{\kappa}_{\tau,t} d(\tilde{x}_t, \tilde{x}_t + \delta) \quad \forall \delta \in \Delta_\infty(\tilde{x}_t)$. Noting that under $\mathcal{E}$ $f(x) \geq \text{lcb}_t(x) \forall x \in \mathcal{X}$, we obtain the first inequality. For the second inequality observe that from the definition of $\bar{\kappa}_{\tau,t}(x)$ we know that $\exists \delta' \in \Delta_\infty(\tilde{x}_t)$ such that $\text{ucb}_t(\tilde{x}_t + \delta') = \tau - \bar{\kappa}_{\tau,t} d(\tilde{x}_t, \tilde{x}_t + \delta')$. Noting that $f(x) \leq \text{ucb}_t(x)$ for all $x$ under $\mathcal{E}$ concludes the proof.

## C.6 Proof of Corollary 3.5

By Lemma 3.4 we have $\Delta_{\underline{\epsilon}_{\tau,t}}(\tilde{x}_t) \subseteq \Delta_{\epsilon_\tau}(\tilde{x}_t)$, then for any $\delta \in \Delta_{\underline{\epsilon}_{\tau,t}}(\tilde{x}_t)$, we have $f(\tilde{x}_t + \delta) \geq \text{lcb}_t(\tilde{x}_t + \delta) \geq \tau$ under the selection rule and the good event $\mathcal{E}$. Conversely again by Lemma 3.4 we have $\Delta_{\epsilon_\tau}(\tilde{x}_t) \subseteq \Delta_{\bar{\epsilon}_{\tau,t}}(\tilde{x}_t)$. Assuming $\mathcal{X}$ is continuous, $\exists \delta' \in \Delta_{\epsilon_\tau}$ such that $f(\tilde{x}_t + \delta') = \tau$. Noting that $\delta' \in \Delta_{\bar{\epsilon}_{\tau,t}}$ completes the proof.

## C.7 Proof of Proposition 3.7

Consider the following instance of our problem. Let $\mathcal{X} = [0,1]$ and $f(x) = x$, noting that this is an element of RKHS with a linear kernel. The Lipschitz constant of this function is 1. Let the satisficing goal $\tau = 1$ which is the function maximum. If $\epsilon_t < 1$, no matter what action is chosen by the learner, the adversary can choose a perturbation $\delta_t = -\epsilon_t$ to obtain a reward $f(x_t + \delta_r) \leq \tau - \epsilon_t$, making the instantaneous lenient regret $r^l = (\tau - f(x_t + \delta_t))^+ \geq \epsilon_t$. This constitute a worst case lower bound, hence the bound is tight and the linear penalty term is not avoidable in general.

## C.8 Proof of Proposition 4.1

WLOG assume that $\exists \delta \in \Delta_\infty(x)$ such that $f(x+\delta) \leq \tau$. Let $\Delta'(x) = \{\delta \in \Delta_\infty(x) \mid f(x+\delta) < \tau\}$. Notice that (10) is then equivalent to

$$\kappa_{\tau,p}(x) := \min k \text{ s.t. } f(x + \delta) \geq \tau - [kd(x, x + \delta)]^p, \quad \forall \delta \in \Delta'(x),$$

since for any $\delta \notin \Delta'(x)$, the inequality constraint already holds for any $k \geq 0$.

Then we can write the equivalent formulation for an action that satisfies $f(x) \geq \tau$

$$\kappa_{\tau,p}(x) := \sup_{\delta \in \Delta'(x)} \frac{[\tau - f(x + \delta)]^{1/p}}{d(x, x + \delta)} .$$

As $p \to \infty$, note that for each fixed $x$ and $\delta \in \Delta'(x)$, $[\tau - f(x + \delta)]^{1/p} \downarrow 1$ monotonically. Hence, by monotone convergence,

$$\kappa_{\tau,p}(x) = \sup_{\delta \in \Delta'(x)} \frac{[\tau - f(x + \delta)]^{1/p}}{d(x, x + \delta)} \downarrow \sup_{\delta \in \Delta'(x)} \frac{1}{d(x, x + \delta)},$$

$$\frac{1}{\kappa_{\tau,p}(x)} = \min_{\delta \in \Delta'(x)} d(x, x + \delta) = \epsilon_\tau(x) .$$

Since $\kappa_{\tau,p}(x)$ decreases pointwise to $1/\epsilon_\tau(x)$ for all $x$, we can exchange the limit and the minimization by uniform convergence:

$$\lim_{p \to \infty} \min_{x \in \mathcal{X}} \kappa_{\tau,p}(x) = \min_{x \in \mathcal{X}} \lim_{p \to \infty} \kappa_{\tau,p}(x) = \min_{x \in \mathcal{X}} \frac{1}{\epsilon_\tau(x)} = \max_{x \in \mathcal{X}} \epsilon_\tau(x) = x^{\text{RS-2}} .$$

### C.9 Proof of Corollary 4.2

The proof is almost identical to the RS regret proof for AdveRS-1, we include it for completeness. Define the instantaneous regret at round $t$ as $r_t^{rs\text{-}g} := \tau - [\kappa_\tau \epsilon_t]^p - f(\tilde{x}_t + \delta_t)$. The cumulative regret is $R_T^{rs\text{-}g} = \sum_{t=1}^{T} (r_t^{rs\text{-}g})^+$. Assume that the good event $\mathcal{E}$ holds. Then the instantaneous regret can be bounded as

$$r_\tau^{\text{rs-g}} \leq \tau - [\kappa_\tau \epsilon_t]^p - \text{ucb}_t(\tilde{x}_t + \delta_t) + 2\beta_t \sigma_t(\tilde{x}_t + \delta_t) \tag{34}$$
$$\leq \tau - [\kappa_\tau \epsilon_t]^p - (\tau - [\overline{\kappa}_{\tau,p,t}(\tilde{x}_t) d(\tilde{x}_t, \tilde{x}_t + \delta_t)]^p) + 2\beta_t \sigma_t(\tilde{x}_t + \delta_t) \tag{35}$$
$$\leq \tau - [\kappa_\tau \epsilon_t]^p - (\tau - [\overline{\kappa}_{\tau,p,t}(\tilde{x}_t) \epsilon_t)]^p + 2\beta_t \sigma_t(\tilde{x}_t + \delta_t) \tag{36}$$
$$= \epsilon_t^p (\overline{\kappa}_{\tau,p,t}^p - \kappa_\tau^p) + 2\beta_t \sigma_t(\tilde{x}_t + \delta_t) \tag{37}$$
$$\leq 2\beta_t \sigma_t(\tilde{x}_t + \delta_t) = 2\beta_t \sigma_t(x_t) . \tag{38}$$

(35) follows from the guarantee of the RS-G formulation $\text{ucb}_t(\tilde{x}_t + \delta) \geq \tau - [\overline{\kappa}_{\tau,p,t} d(\tilde{x}_t, \tilde{x}_t + \delta)]^p$ for any $\delta \in \Delta_\infty(\tilde{x}_t)$. (37) follows from $\overline{\kappa}_{\tau,p,t} \leq \kappa_{\tau,p,t}$ since $\text{ucb}_t(x) \geq f(x)$ for all $x$. Then we bound the cumulative robust satisficing regret, following the same standard steps to obtain an upper bound of:

$$R_T^{rs\text{-}g} \leq 4\beta_T \sqrt{T\gamma_T} .$$

$\square$

## D  Implementation details of the algorithms

We note that maximizing UCB and LCB over a compact domain is non-trivial due to the non-convex nature of the posterior mean and variance. Like foundational works (*e.g.*, [34]), we focus on theoretical guarantees and assume an oracle optimizer, abstracting computational details. This challenge is common across all GP-based methods, including the RO algorithms we compare against, and in practical BO, various optimization (*e.g.*, gradient descent or quasi-Newton) techniques are used to address it.

In our experiments, we work with discretized domains and the implementation of the acquisition functions of our algorithms have complexity $\mathcal{O}(N^2)$ where $N = |\mathcal{X}|$. Notably, this complexity is the same as that of RO-based algorithms. Specifically, in all algorithms that we implement (RS or RO based), action-specific calculations are performed in an inner loop, while an outer loop is used the selection of best action.

## E  Additional Experimental Results

### E.1  Inverted Pendulum Experiment

In this experiment, we focus on optimizing a parameterized controller to achieve robust performance in controlling an inverted pendulum. The state vector $s_k$ consists of four variables: cart position, pendulum angle, and their derivatives. The simulation is conducted using OpenAI's Gym environment.

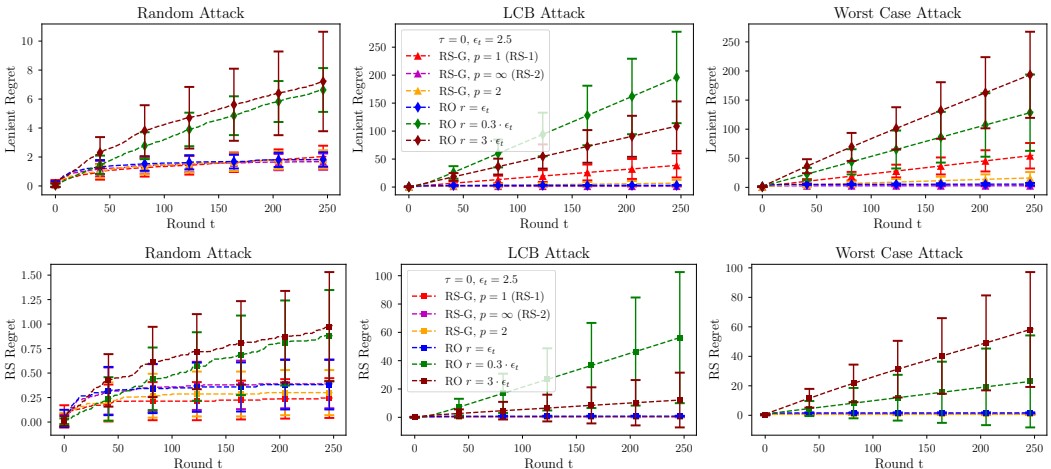

Figure 7: Lenient and RS regret results of the inverted pendulum experiment. Perturbation budget is $\epsilon_t = 2.5$ and $\tau = 0$ for all $t$. STABLEOPT run with: $r = \epsilon_t$, $r = 3\epsilon_t$, and $r = 0.3\epsilon_t$. Attack schemes are indicated in the supertitles.

The system dynamics are described by $\boldsymbol{s}_{k+1} = h(\boldsymbol{s}_k, u_k)$, where $\boldsymbol{s}_k$ is the 4-dimensional state vector, and $u_k \in [-10, 10]$ is the control action representing the voltage applied to the controller at step $k$. We implement a static feedback controller similar to the one in [43], defined as $u_k = \boldsymbol{F}\boldsymbol{s}_k$, where $\boldsymbol{F} \in \mathbb{R}^{1 \times 4}$ is the gain matrix. We structure $\boldsymbol{F}$ as a Linear Quadratic Regulator (LQR) using the linearized system dynamics $(\boldsymbol{A}, \boldsymbol{B})$ around the equilibrium point. The parameterization, which has been shown to be learnable by GP's [43] is given by $\boldsymbol{F} = \text{dlqr}(\boldsymbol{A}, \boldsymbol{B}, \boldsymbol{W}_s(\boldsymbol{\theta}), \boldsymbol{W}_u(\boldsymbol{\theta}))$ with the following parameterization:

$$\boldsymbol{W}_s(\boldsymbol{\theta}) = \text{diag}(10^{\theta_1}, 10^{\theta_2}, 10^{\theta_3}, 0.1), \qquad \theta_{1,2,3} \in [-3, 2],$$
$$\boldsymbol{W}_u(\boldsymbol{\theta}) = 10^{-\theta_4}, \qquad \theta_4 \in [1, 5].$$

We discretize the parameter space into 4096 points and compute the cost for each parameter by simulating it over 2000 seconds, using the same cost function as in [44]. After the simulation, the cost function is standardized and clipped to the range $[-2, 2]$. For the GP kernel, we use a Radial Basis Function (RBF) kernel with Automatic Relevance Determination (ARD), with hyperparameters selected using 400 samples prior to the experiment.

For adversarial perturbations, we consider an adversary that damages the learning process by perturbing the selected parameter. The unknown perturbation budget is constant with $\epsilon_t = 2$. The satisficing goal is set to $\tau = 0$ which corresponds to $\sim 25$ percentile of the objective function. Figure 7 shows the regrets of the algorithms over a time horizon of $T = 250$. AdveRS family of algorithms matches the STABLEOPT that has the perfect knowledge of the adversarial budget, and they outperform STABLEOPT otherwise. While AdveRS-1 achieves linear regret, it can still perform better than STABLEOPT when the perturbation budget is misspecified.

### E.2 Misspecified Kernel

In this experiment, we evaluate the performance of our algorithms when the GP is run with a kernel that is not aligned with the true function. Specifically, we use the function shown in Figure 2, but unlike Experiment 1, we replace the polynomial kernel with an RBF kernel, using a lengthscale of $0.1$ and a variance of $10$. All algorithms are run with the same parameters as in Experiment 1 with $\tau = -10$ and $\epsilon_t = 0.5$. Figure 8 shows that AdveRS-2 maintains strong performance even when the kernel is misspecified, while AdveRS-1 outperforms STABLEOPT when the ambiguity ball is not correctly estimated. Figure 9 shows the RS regrets and we see that our algorithms achieve sublinear results, while STABLEOPT with misspecified ambiguity radius can achieve linear RS regret. All plots show the mean results over 100 independent runs with error bars showing std/2.

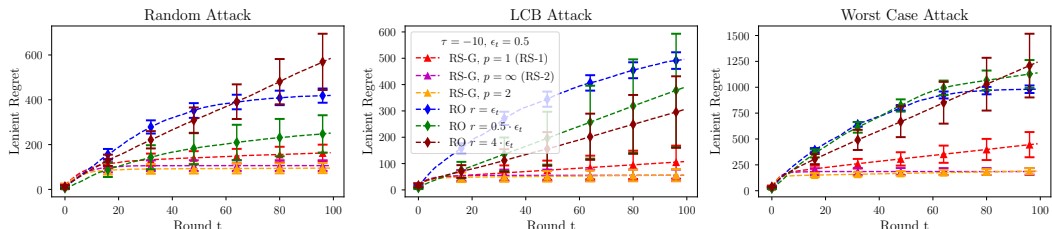

Figure 8: Lenient regret results for synthetic experiment using a misspecified kernel, with a perturbation budget of $\epsilon_t = 0.5$ and $\tau = -10$ for all $t$. STABLEOPT run with: $r = \epsilon_t$, $r = 4\epsilon_t$, and $r = 0.5\epsilon_t$.

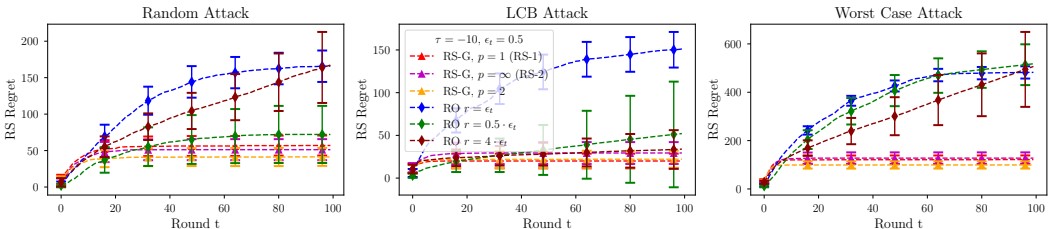

Figure 9: Robust satisficing regret results for synthetic experiment using a misspecified kernel, with a perturbation budget of $\epsilon_t = 0.5$ and $\tau = -10$ for all $t$. STABLEOPT run with: $r = \epsilon_t$, $r = 4\epsilon_t$, and $r = 0.5\epsilon_t$.

### E.3 Robust Satisficing with Thompson sampling

**Thompson Sampling.** Thompson sampling [45, 46] selects the next point by sampling a function $\tilde{f}$ from the GP posterior and then choosing

$$x_t \in \arg\max_{x \in \mathcal{X}} \tilde{f}(x).$$

This randomized policy implicitly balances exploration and exploitation by occasionally sampling from regions of high uncertainty.

---

**Algorithm 3** AdveRS-G-TS

---

1: **Input:** Kernel function $k$, $\mathcal{X}$, $\tau$, $p$, time horizon $T$
2: **Initialize:** $\mathcal{D}_0 = \emptyset$ (empty dataset), and $\mu_0(x) = 0$, $\sigma_0(x) = 1$ $\forall x \in \mathcal{X}$
3: **for** $t = 1$ to $T$ **do**
4:     Sample $\tilde{f}_t(x) \sim \mathcal{GP}\big(\mu_{t-1}(x), \sigma_{t-1}^2(x)\big), \forall x \in \mathcal{X}$
5:     Compute $\overline{\kappa}_{\tau,p,t}(x)$ using (39), $\forall x \in \mathcal{X}$
6:     Select $\tilde{x}_t = \arg\min_{x \in \mathcal{X}} \overline{\kappa}_{\tau,t}(x)$
7:     Adversary selects perturbation $\delta_t \in \Delta_{\epsilon_t}(\tilde{x}_t)$
8:     Sample $x_t = \tilde{x}_t + \delta_t$, observe $y_t = f(x_t) + \eta_t$
9:     $\mathcal{D}_t = \mathcal{D}_{t-1} \cup \big\{(x_t, y_t)\big\}$
10:     Update GP posterior using (5)
11: **end for**

---

**Algorithm for RS-G using Thompson sampling.** In order to perform Bayesian optimization with the objective as RS-G, building on the algorithm family of AdveRS, we propose *Adversarially Robust Satisficing-General-Thompson Sampling* (AdveRS-G-TS) algorithm (see Algorithm 3).

At the beginning of each round $t$, AdveRS-G-TS samples a random representative $\tilde{f}_t$ from the posterior distribution of GP. Then, replacing the $f$ in (10) instead with $\tilde{f}_t$ and rearranging as in , it

computes the Thompson $p$-fragility defined as:

$$\overline{\kappa}_{\tau,p,t}(x) := \left( \max_{\delta \in \Delta_\infty(x) \setminus 0} \frac{\tau - \tilde{f}_t(x+\delta)^{1/p}}{d(x, x+\delta)} \right)^+ , \tag{39}$$

if $\tilde{f}_t(x) \geq \tau$, and otherwise $\overline{\kappa}_{\tau,p,t}(x) := \infty$. For each action, the Thompson $p$-fragility $\overline{\kappa}_{\tau,p,t}(x)$ provides a random sample of the true fragility $\kappa_{\tau,p}(x)$. AdveRS-G-TS then selects the action with the smallest random fragility, $\tilde{x}_t = \arg\min_{x \in \mathcal{X}} \overline{\kappa}_{\tau,p,t}(x)$, to guide exploration and exploitation.

**Insulin dosage experiment using AdveRS-G-TS**

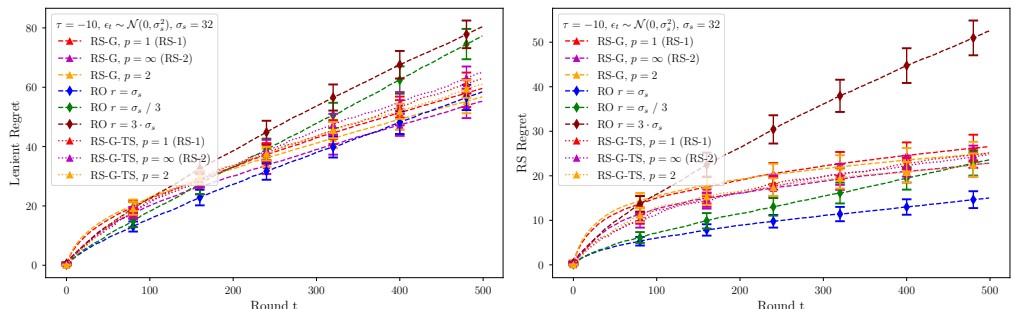

Figure 10: Lenient and RS regret results of insulin dosage experiment. Note that the results for algorithms other than AdveRS-G-TS are the same as in 10 and are averaged over 100 random runs, while AdveRS-G-TS results are averaged over 20 runs.

### E.4 Effect of $\tau$ and $r$

Table 2 shows the varying effects of the parameters. The lenient regret of all algorithms are calculated on the synthetic function from Experiment 1. The perturbations are sampled i.i.d. from a normal distribution $\mathcal{N}(0, 0.3^2)$. For this experiment, we report results from 20 random runs. Results show that the target oriented nature of RS approaches makes them better suited for the satisficing objective under unknown perturbations.

| Alg. \ $\tau$ | $-30.00$ | $-20.00$ | $-10.00$ | $0.00$ | $10.00$ |
|---|---|---|---|---|---|
| RS1 | 103.83 | 201.97 | 287.81 | 651.38 | 1478.62 |
| RS2 | **47.09** | **104.01** | 185.39 | **434.30** | **626.48** |
| RSG | 52.70 | 113.03 | **177.90** | 619.17 | 1505.98 |
| RO, r=0.10 | 112.29 | 207.81 | 335.10 | 458.96 | 645.07 |
| RO, r=0.57 | 94.20 | 180.79 | 293.17 | 537.80 | 1489.18 |
| RO, r=1.05 | 74.30 | 138.19 | 224.45 | 723.06 | 1695.44 |
| RO, r=1.52 | 55.05 | 126.99 | 258.48 | 1053.13 | 2121.94 |
| RO, r=2.00 | 60.05 | 284.83 | 722.58 | 1523.14 | 2721.97 |

Table 2: Average lenient regrets for synthetic experiment for RS-1, RS-2 and RO algorithms with varying $\tau$ and $r$ values. Best values are in bold for each $\tau$ value.

