# OpenReview forum: "Robust Satisficing Gaussian Process Bandits Under Adversarial Attacks"
_NeurIPS.cc/2025/Conference — NeurIPS 2025 poster_

### Official Review · Reviewer_vqo3 · 2025-06-29

**Clarity:** 4
**Significance:** 2
**Originality:** 2
**Rating:** 5
**Confidence:** 4

**Summary:**

This paper introduces a novel framework for robust satisficing (RS) in Gaussian Process bandit optimization, where the goal is to reliably achieve a threshold performance level under unknown adversarial perturbations. Unlike traditional robust optimization, which focuses on worst-case outcomes, the authors propose two RS-based formulations (RS-1 and RS-2) that ensure satisfactory performance across a range of perturbations without requiring prior knowledge of the adversary’s strength. They develop algorithms (AdveRS-1 and AdveRS-2) with provable regret bounds and introduce a general RS-G framework that interpolates between RS-1 and RS-2 via a tunable parameter. Empirical results show that these methods outperform existing robust optimization approaches.

**Questions:**

Please see the listed strengths and weaknesses.
Although, I have no specific questions.

**Ethical Concerns:**

["NO or VERY MINOR ethics concerns only"]

**Final Justification:**

I have been positive about this paper from the beginning and find it interesting, but I am maintaining my positive score because I remain unconvinced about its potential for high impact.

**Limitations:**

Yes, some limitations are acknowledged at the end of the paper. I believe this work shares similar limitations with previous studies, but they are not critical.

**Paper Formatting Concerns:**

The paper is well-written. No formatting concerns.

**Quality:**

3

**Strengths And Weaknesses:**

Strengths:
* Clear and well-written paper with helpful illustrations.
* Related work is well-situated and comprehensive.
* The paper clearly identifies a setting gap: combining robust objectives, lenient regret, and input perturbations during interaction (beyond standard reported point-only attacks). The algorithm design is also novel.
* Proposed algorithms make a solid contribution to the robust GP bandits literature.
* Introduces novel objectives grounded in robust optimization theory.
* Provides sub-linear regret guarantees for the proposed setting.

No major weaknesses were identified.
One could argue that the motivation for choosing these specific objectives and the interaction protocol could be elaborated further. Additionally, the regret analysis largely build upon prior work.
However, I do not consider these points to be significant weaknesses of the paper.
The insulin dosage experiment could be used to further illustrate and motivate the defined interaction protocol, as well as the robust objective and the threshold τ.

My rating falls between 4 and 5. In my view, there is no obvious reason to reject this paper (will rethink this after I read other reviews); however, some may find it less useful, less motivated, or lacking in novelty, though I consider this a subjective judgment. Overall, the paper is solid and makes meaningful contributions. I find it particularly clever how the authors identified a gap in the literature, namely, the combination of lenient regret, input attacks, and robust objectives. To the best of my knowledge, this setting has not been explored before.

---

> ### Author Rebuttal · Authors · 2025-07-30
>
> Dear Reviewer vqo3,
>
> Thank you for your feedback. We greatly appreciate your positive assessment of our work's contributions.
>
> We agree that the insulin dosage experiment could be used to further illustrate and motivate the problem formulation and the robust satisficing objective. Following your feedback, we will mention this experiment earlier in the paper to set a better context for the rest of our paper. We would be eager to address any questions you might have following the initial reviews and our rebuttals.

---

### Official Review · Reviewer_SL7m · 2025-07-01

**Clarity:** 3
**Significance:** 3
**Originality:** 3
**Rating:** 5
**Confidence:** 3

**Summary:**

This paper proposes a Bayesian optimization method for efficiently identifying robust solutions under adversarial perturbations in the optimization problem of expensive-to-evaluate black-box functions. Representative robustness measures such as worst-case measures have the problem of selecting overly conservative solutions depending on the parameters that determine the range of perturbations. In contrast, this paper considers robust satisficing, a moderate robustness measure. First, two definitions of robust satisficing, RS-1 and RS-2, are given. RS-1 is defined as the minimum value of slope at which the threshold can be reduced in linear order according to the magnitude of the perturbation, while RS-2 is defined as the maximum magnitude of the perturbation that causes the threshold to be violated. Furthermore, by generalizing the linearity in RS-1, the authors propose RS-G, a general robust satisficing that connects RS-1 and RS-2. In the proposed method, an acquisition function based on GP-UCB is given for decision-making, and theoretical bounds for two types of regrets, $R^l_T$ and $R^{rs}_T$, are provided. Due to the presence of adversaries, the standard regret $R^l_T$ is not sublinear, while the proposed method achieves a sublinear regret bound for $R^{rs}_T$, which accounts for the degree of perturbation. Numerical experiments are conducted to compare the performance with comparison methods.

**Questions:**

This paper is well written. I believe that Weaknesses 2 and 3 are issues for future consideration and do not diminish the value of this paper. However, Weakness 1 requires further explanation. Furthermore, the content of this paper can be improved by clarifying the following unclear points:

1. Among the various robustness measures, what are the advantages of robust satisficing? Existing studies [30] and [31] consider VaR and CVaR, which are threshold-based robustness measures. Additionally, PTR is also a threshold-based measure, as it represents the probability of exceeding the threshold, and thus resembles the robustness measures discussed in this paper. Additionally, distributionally robust measure [27] is an intermediate robustness measure between worst-case and expectation measures. Considering these points, a detailed explanation of the specific advantages of robust satisficing is necessary.

2. Can the RKHS assumption be changed? As in the continuous setting of Srinivas et al. [34], if we assume probabilistic Lipschitz continuity as in Theorem 2 of [34], and normality of the observation noise, it seems possible to adapt the Bayesian setting.

3. Can this method be applied in a situation where only $\tilde{x}_t$ is known, and the values of the adversarial perturbations $\delta_t$ and the actual inputs $x_t$ are unknown? Is there any possibility of extending this method to such a setting?

4. In the footnote on page 3, it is stated that Lipschitz continuity is not necessarily required for $f$, but in line 230, Lipschitz continuity is used. Is Lipschitz continuity ultimately required?

5. In equation (2), is $k$ allowed to take negative values?

**Ethical Concerns:**

["NO or VERY MINOR ethics concerns only"]

**Final Justification:**

The author's rebuttal cleared up my confusion, but it was not enough to raise my rating to 6. Furthermore, the most significant limitation of this paper, namely, the assumption that the adversarial perturbations are known, still remains. However, while I agree this as a limitation, I do not consider it a fatal flaw. The reason is that constructing a reasonable model without knowing the values of adversarial perturbations is inherently a challenging problem, and thus this issue should be addressed in future work. Hence, I believe this limitation should not lower the evaluation of this paper. Therefore, I have not changed my initial score of 5.

**Limitations:**

Yes. These are stated in Conclusion. However, in my understanding, this paper assumes that the values of the adversarial perturbations and the exact inputs after perturbation are known. Therefore, I believe that the limitation should also include the fact that the paper does not address cases where the values of the adversarial perturbations are unknown. On the other hand, the main focus of this paper is the development of a methodological framework and theoretical analysis for Bayesian optimization of robustness measure under adversarial perturbations, so it does not fall under the category of potential negative social impact.

**Quality:**

3

**Strengths And Weaknesses:**

The strengths of this paper are as follows:

1. It deals with Bayesian optimization problems under adversarial perturbations using RS-1, RS-2, and their generalizations, RS-G, which are robustness measures that are not overly conservative.

2. The authors designed an acquisition function based on GP-UCB and demonstrated that a theoretically sub-linear regret bound can be achieved for regret that accounts for the influence of perturbations, even under adversarial perturbations.

3. Numerical experiments compare the robustness of the proposed method with existing methods under various perturbation conditions, demonstrating its effectiveness.

On the other hand, the weaknesses of this paper are as follows:

1. The comparison with related robustness measures is insufficient. The distributionally robust measure [27] is also moderate robustness measure. Furthermore, while RS-1, RS-2 and RS-G are threshold-based robustness measures, value-at-risk (VaR) [30], conditional VaR (CVaR) [31], and probability threshold robustness measures (PTR), $PTR= \mathbb{P} _{\delta} ( f( x + \delta ) > \tau)$, are also  threshold-based measures. It is necessary to explain these characteristics and the differences from the proposed method in more detail.

(PTR): Iwazaki et al., Bayesian Quadrature Optimization for Probability Threshold Robustness Measure, Neural Computation 33 (12), 2021

2. The robustness of robust satisficing strongly depends on the given threshold $\tau$. As mentioned in  Conclusion, depending on the value of $\tau$, overly conservative or optimistic solutions may be obtained, and the problem remains of how to deal with situations where domain knowledge is insufficient and it is not possible to design a reasonable $\tau$.

3. The value of the adversarial perturbation is assumed to be known. In the proposed method, the surrogate model is calculated using $(x_t,y_t)$ under the assumption that the value of the perturbation $\delta_t$ given by the adversary and the value of $x_t$ including the perturbation are known for the input $\tilde{x}_t$ given in advance. On the other hand, in actual situations, if the adversary secretly provides the perturbation $\delta_t$ and the user receives $x_t = \tilde{x}_t + \delta_t$ without knowing the situation, the user only knows $(\tilde{x}_t,y_t)$, so the problem remains that the surrogate model calculation using $(x_t,y_t)$ cannot be performed.

[Quality: 3]
The proposed method performs theoretical analysis of regret under the RKHS setting based on GP-UCB theory. Numerical experiments confirm the practical performance of the proposed method under various adversarial perturbations. The paper also explains the theoretical limitations of the problem setting, namely that the regret bound becomes linear for standard  regret due to the existence of adversarial perturbations.  On the other hand, it appropriately explains that the upper bound of the regret including adversarial perturbations becomes sublinear. Furthermore, it mentions that setting the threshold $\tau$ in cases where little prior information is available is a problem as a limitation and possibility for future development of this method.

[Clarity: 3]
This paper appropriately explains the problem setting, assumptions and results of the theorems, and their interpretations. Furthermore, in the definition of robust satisficing, the paper uses figures as well as mathematical formulas to visually explain the meaning of these robustness measures, making them easier to understand.

[Significance: 3]
This paper provides ideas for designing practical methodologies for BO for robustness measures in the presence of adversarial perturbations. It can be expected to derive the regret bounds under adversarial perturbations for other robustness measures in a manner similar to this paper. Additionally, unlike overly conservative robustness measures such as worst-case measures, this paper highlights the importance of considering robustness measures like robust satisficing, which strike a moderate balance.

[Originality: 3]
It has been shown that existing GP-UCB for maximization problems can be extended to acquisition functions for robust satisficing under adversarial perturbations. Furthermore, as a point that can advance the field of robust BO, a new methodology for performing regret analysis has been provided by considering robust satisficing as a moderate measure. This enables the exploration of extensions to multi-objective optimization using multiple robust satisficing measures. In this sense, this research is highly original.

Based on the above, I have evaluated this paper as 5, as it meets all the criteria for acceptance.

---

> ### Author Rebuttal · Authors · 2025-07-30
>
> Dear Reviewer SL7m,
>
> Thank you for your feedback. We greatly appreciate your positive assessment of our work's quality, clarity, significance, and originality. Below, we address the weaknesses and questions you raised.
>
> **Weakness:**
>
> (1. The comparison with related robustness measures is...)
> We thank the reviewer for their suggestion and agree that comparison with related robustness measures is relevant. But we also note that this would require some change in the problem formulation, as currently we do not model the perturbation as a random variable. The measures the reviewer mentions, such as VaR, CVaR, and DRO, are probabilistic measures. For direct comparison, the problem formulation could be altered to contain a random covariate in the objective function, i.e., $f(x, Z)$, instead of just considering $f(x)$. Then, depending on whether the distribution of $Z$ is known or not, one can propose variations of our RS measures tailored to the problem formulation. One possibility when the distribution of $Z$ is known is to consider the robustness measure:
> $$
>     \kappa\_{\tau,p}(x) = \inf\_{k \geq 0} k \quad \text{s.t.} \quad \mathbb{P}\_Z \left( f(x,Z) \geq \tau - [k  d(Z, z\_\text{ref})]^p \right) \geq 1 - \delta~,
> $$
> where $z\_\text{ref}$ is the best estimate of the random covariate. We further note the assumption that input perturbations follow a distribution would limit the power of the adversary. Since the setup in our paper sets no limitation on the adversary, it is more general. We will emphasize these differences in the revised version of the paper.
>
> (2. The robustness of robust satisficing strongly depends on...) Indeed, the performance of our algorithms depend on the satisficing threshold $\tau$. In general, achieving good lenient regret for larger $\tau$ values is more difficult. However, we argue that selecting $\tau$ is much more interpretable and straightforward compared to selecting the ambiguity ball radius $r$ for the RO approach. The insulin dosage experiment also serves as an example to show how a domain expert might approach setting $\tau$ in a real-world setting.
>
> (3. The value of the adversarial perturbation is assumed to be known...)
> This is a very good point, one which is raised by other reviewers as well. There is work on GPs where the input is not directly observed. For example, to account for input uncertainty in the surrogate model, [A] (reference no 23 in the paper) assumes that perturbations follow a distribution and defines a GP over probability distributions. Further, they assume access to a reference distribution estimating the true query location. As our setting does not have any such assumptions on the adversary, adapting it to uncertain inputs remains an open problem. We will mention this as an important future research direction.
>
> [A] Rafael Oliveira, Lionel Ott, and Fabio Ramos. Bayesian optimisation under uncertain inputs. In The 22nd international conference on artificial intelligence and statistics, pages 1177–1184. PMLR, 2019.
>
> **Questions:**
> - (1. Among the various robustness measures...) Following our discussion from the weakness section, while VaR and CVaR are threshold-based robustness measures, they measure mainly the magnitude of the tail-end of outcomes. VaR, for example, measures the level of loss that will not be exceeded with probability $1 - \alpha$. The proposed RS measures are different in the sense that they also give consideration to likely outcomes. For example, the proposed measures require an action to achieve $\tau$ when there is no perturbation, and be close to $\tau$ when the perturbation is small. This separates the RS measures from VaR and CVaR.
> - (2. Can the RKHS assumption be changed...) It is indeed possible to adapt our results to the Bayesian setting. Since one can come up with high probability upper and lower confidence bounds in the Bayesian setting as well, the analysis follows.
> - (3. Can this method be applied in a situation...) As we argued in the weakness section, it is possible to extend our methods to such a setting by incorporating input uncertainty-aware surrogate models. However, this will probably require a more restricted adversary, and remains as an area of future work.
> - (4. In the footnote on page 3...) It is true that we use the Lipschitz continuity of $f$ with respect to the kernel metric in our analysis. The footnote on page 3 states that $f$ is not necessarily Lipschitz with respect to any metric, but it is Lipschitz with respect to the kernel metric, and that we assume $d(\cdot, \cdot)$ is the kernel metric moving forward.
> - (5. In equation (2)...) Yes, we allow $k$ to have negative values. However, notice that the fragility being negative means that the objective function lies entirely above the satisficing threshold, i.e., $f(x) > \tau,$ $\forall x\in{\cal X}$, which implies that the learning problem becomes trivial. In practical cases, where adversarial perturbations can drive the system towards values less than the threshold, we do not observe negative $k$.
>
> **Limitations:** We agree with the reviewer that a limitation of our work is the assumption that adversarial perturbations are observable. In the revised version, we will mention this as a potential direction for future work.

---

> > ### Comment · Reviewer_SL7m · 2025-08-01
> >
> > Thank you for your response. I had misunderstood the assumption regarding perturbations, but it has been clarified. I have also resolved my other questions. Moreover, as the authors mentioned, I agree that assuming perturbations are known is a limitation, and that additional assumptions are necessary to construct a valid model without this assumption. Therefore, I will maintain my original score.

---

### Official Review · Reviewer_X7S6 · 2025-07-02

**Clarity:** 3
**Significance:** 2
**Originality:** 3
**Rating:** 3
**Confidence:** 3

**Summary:**

This paper aims to solve the robust satisficing problem under adversarial attack. The authors first define two different formulations: RS1 and RS2, and extends the UCB/LCB to the corresponding optimistic/pessimistic bounds, followed by a unified algorithm.

**Questions:**

1. Could you provide some real-world cases for the RS1 and RS2 formulations?
2. In the experiment, what if the attacker always return the same perturbed point within a budget? Will this attack affect the performance?
3. Can the critical radius/action fragility extend to other existing acquicition functions, e.g., EI or PI?

**Ethical Concerns:**

["NO or VERY MINOR ethics concerns only"]

**Final Justification:**

The authors' rebuttal addresses most of my concerns, especially about the information the learner can observe. Therefore, I will increase my score to 4 (borderline accept)

**Limitations:**

yes

**Paper Formatting Concerns:**

No formatting concerns

**Quality:**

2

**Strengths And Weaknesses:**

### Strength
1. In general, this paper is well-presented and easy to follow.
2. The robust satisficing problem can be quite important and applicable to many real-world cases.
3. Solid experiment.

### Weakness
- In the current setting, the authors assume the learner can explicitly observe the perturbed point $x_t$ (Line 89) and develop their algorithm based on this assumption. However, this could be infeasible for many real-world cases. As pointed out in [1,2] the perturbed value cannot be observed.
- Also, if the pertubed value is explicitly observed, it is straightforward to learn a GP without any ambiguity, making the decision much easier.
- The key design of acquisition function, i.e., the extensions of UCB/LCB to optimistic/pessimistic fragility, as well as the optimistic critical radius, are quite intuitive.

[1] Yang et al., "Efficient robust Bayesian optimization for arbitrary uncertain inputs", NeurIPS 2023

[2] Oliveira et al., "Bayesian optimisation under uncertain inputs", AISTATS 2019

---

> ### Author Rebuttal · Authors · 2025-07-30
>
> Dear Reviewer X7S6,
>
> Thank you for your feedback. We greatly appreciate your acknowledgement of the strengths of our work and your thoughtful criticisms. Below, we respond to each comment in detail. If you have any further questions, we are eager to engage in a more detailed discussion.
>
>
> **Weakness:**
>
> As the reviewer highlights, there is work on GP's where the inputs are not directly observed. [1] incorporates input uncertainty into the posterior using an MMD based method, and [2] (reference no 23 in the paper) defines a GP over probability distributions. To account for input uncertainty in the surrogate model, however, both approaches rely on additional assumptions about the nature of perturbations. In particular, both [1] and [2] assume that perturbations follow a distribution. Further, [1] requires that the learner can sample from the perturbation distribution, whereas [2] assumes access to a reference distribution that approximates the true query location.
> In our setting, no such assumptions are made, making the uncertain input formulation inherently more challenging. Even if the adversary samples the perturbation from a distribution, they may change this distribution from round to round. Without additional constraints on the adversary, it's not possible to reliably estimate the perturbation distribution, and construct an uncertainty aware surrogate model. That being said, after introducing additional assumptions about the adversary's behaviour, we believe our robust satisficing framework could be extended to explicitly account for input uncertainty, which would make an interesting direction for future research. We appreciate the reviewer pointing out this related line of work. We will discuss its relation to our work in detail in the revised paper.
>
>
> **Questions:**
>
> - (...real-world cases for RS...) Apart from the insulin dosage experiment in the paper, both RS-1 and RS-2, as well as RS-G, can be readily applied to many real-world problems where RO is already used. Some examples include:
>     - The real-world robot control tasks given in [1] and [2] fit into our model with a slight environmental twist. Consider the scenario (for example in space missions) where the communication with the robot is discrete. An autonomous drone is sent to location $x_t$ to perform a measurement task, but due to fuzzy control or environmental necessities, ends up at location $\tilde{x}_t$. The robot can share $\tilde{x}_t$ along with the measurement it took when the communication is available (e.g. at the base).
>     - One recent promising use of RO (DRO in particular) is for preference alignment in LLM's [A]. The authors tackle the problem of distribution shifts due to synthetic data in LLM preference alignment using a DRO based method. One challenge which the authors mention in the paper is that DRO can suffer from over-pessimism, which manifests as poor generalization. An RS based approach can possibly be a strong alternative and a promising future research avenue.
>
>     Apart from these, portfolio optimization and inventory and supply chain management are among the domains where RO has been extensively studied [B]. Both RO and RS have also been used in Markov decision processes (MDPs) with uncertainty present [C, D, E]. Further, [F] presents several examples where RS-1 can be used, such as portfolio optimization and the knapsack problem.
>
>     [A] Zhu, Mingye, et al. "Leveraging Robust Optimization for LLM Alignment under Distribution Shifts." arXiv preprint arXiv:2504.05831 (2025).
>
>     [B] Dimitris Bertsimas, David B Brown, and Constantine Caramanis. Theory and applications of robust optimization. SIAM review, 53(3):464–501, 2011.
>
>     [C] Huan Xu and Shie Mannor. Distributionally robust markov decision processes. Advances in Neural Information Processing Systems, 23, 2010.
>
>     [D] Xiaoting Ji, Yifeng Niu, and Lincheng Shen. Robust satisficing decision making for unmanned aerial vehicle complex missions under severe uncertainty. PloS one, 11(11):e0166448, 2016.
>
>     [E] Haolin Ruan, Siyu Zhou, Zhi Chen, and Chin Pang Ho. Robust satisficing mdps. In International Conference on Machine Learning, pages 29232–29258. PMLR, 2023.
>
>     [F] Daniel Zhuoyu Long, Melvyn Sim, and Minglong Zhou. Robust satisficing. Operations Research, 71(1):61–82, 2023.
>
> - (...return same perturbed point...) The point attacker returns depends on both the perturbation budget of the attacker, and the point chosen by the learner. In general, the adversary woul not be able to return the same perturbed point for every point the learner chooses due to budget limit. If the attacker can force the same point no matter what point the learner choose, this means the adversary is too strong and, indeed, learning the objective function is impossible.
> However, we underscore that our results do not depend on the attacking scheme of the adversary.
> - (...extend to other acquisition functions...) While extending the notion of critical radius and the fragility to EI or PI would not be straightforward, they can be more readily adapted to other known acquisition functions. For instance, rather than employing an upper confidence bound (UCB) strategy, one could utilize Thompson Sampling to estimate the fragility and critical radius of each action by drawing sample functions from the Gaussian process (GP) posterior. We explored this alternative and obtained encouraging preliminary results. In particular, we extended and reran the insulin dosage experiment presented in the paper to include Thompson Sampling variants of RS-1 and RS-2, denoted RS1T and RS2T, respectively. Below, you can see the lenient regrets across timesteps as a table. As in the main experiments, the results are averaged over 100 runs, with error bars indicating half the standard deviation (std/2). The results show that lenient regret of TS-based RS algorithms are very similar to that of AdveRS-1 and AdverS-2.
>
>     | Algorithm               | t=100        | t=200        | t=300        | t=400        | t=500        |
>     |-------------------------|--------------|--------------|--------------|--------------|--------------|
>     | RS1                     | 25.27 ± 2.06 | 36.38 ± 2.85 | 46.19 ± 3.67 | 54.24 ± 4.35 | 62.35 ± 4.55 |
>     | RS1T                    | 20.59 ± 2.07 | 33.18 ± 2.84 | 43.52 ± 3.37 | 53.05 ± 4.34 | 62.35 ± 4.95 |
>     | RS2                     | 22.83 ± 2.10 | 34.69 ± 2.65 | 44.00 ± 2.96 | 52.69 ± 3.62 | 61.41 ± 4.24 |
>     | RS2T                    | 20.93 ± 2.51 | 34.43 ± 3.17 | 45.54 ± 3.73 | 55.47 ± 4.24 | 65.64 ± 4.76 |
>     | RO $r=\sigma_s$         | 16.96 ± 2.06 | 29.34 ± 2.72 | 40.79 ± 3.46 | 51.47 ± 4.07 | 62.50 ± 4.67 |
>     | RO $r=\sigma_s / 3$     | 20.28 ± 2.23 | 36.74 ± 3.12 | 51.59 ± 3.76 | 66.26 ± 4.61 | 81.27 ± 4.93 |
>     | RO $r=3 \cdot \sigma_s$ | 24.95 ± 2.28 | 42.42 ± 3.47 | 57.98 ± 4.26 | 71.93 ± 4.44 | 86.17 ± 5.44 |

---

> > ### Comment · Reviewer_X7S6 · 2025-08-03
> >
> > Thanks for the detailed reply; it addresses my concerns, so I would like to increase my score by 1 point.

---

### Official Review · Reviewer_wsUa · 2025-07-03

**Clarity:** 4
**Significance:** 4
**Originality:** 3
**Rating:** 5
**Confidence:** 3

**Summary:**

This paper tackles GP optimization with unknown adversarial perturbations by replacing the ambiguity set based RO with an RS framework. The authors propose two algorithms AdveRS-1 and AdveRS-2, corresponding to RS formulations RS-1 and RS-2 based on fragility and critical radius, respectively. They also provide a unified framework controlled by a power parameter. They show that AdveRS-1 delivers a linear lenient regret bound and a sublinear robust satisficing regret, while AdveRS-2 attains sublinear bounds for both with an attainable threshold assumption. Experiments demonstrate that RS-based methods outperform standard RO, particularly when the ambiguity set is misspecified.

**Questions:**

- In all experiments, the satisficing threshold is set to -10. Can the authors justify this choice, and offer guidance on how it should relate to the problem’s reward scale?
- I understand selection of the power $p$ is proposed as future work in the Conclusion, but do the authors have any concrete strategies or preliminary results on how to tune $p$ as a hyperparameter or choosing adaptively at runtime?
- Could the authors include a more detailed discussion of computational cost, possibly both in terms of theoretical complexity and empirical runtime?

**Ethical Concerns:**

["NO or VERY MINOR ethics concerns only"]

**Final Justification:**

I advocate the acceptance of this paper, and I am keeping my score on 5 (accept).

**Limitations:**

The authors adequately addressed the limitations.

**Paper Formatting Concerns:**

No formatting concern.

**Quality:**

3

**Strengths And Weaknesses:**

Strengths
- The paper is well written and thoughtfully structured.
- Each framework and its corresponding proposed algorithm are presented clearly and concisely, with intuitive explanations of when and why they work.
- The assumptions required to improve the theoretical bounds are explicitly stated and seem reasonable.
- Experiments demonstrate meaningful improvements over standard RO baselines and clearly illustrate in which regimes each RS method excels.

Weaknesses
- Guidance on selecting the satisficing threshold is limited. Although as the authors continuosly mention, domain-specific tuning is expected, practical heuristics or rules of thumb are missing. Since performance hinges on whether the threshold is attainable, for fair comparison against RO with misspecified ambiguity set, additional sensitivity analyses especially would strengthen the paper.

Overall, I believe it is a well-written paper that introduces a strong framework and methodology and effectively highlights its advantages.

---

> ### Author Rebuttal · Authors · 2025-07-30
>
> Dear Reviewer wsUa,
>
> Thank you for your feedback. We're grateful for your acknowledgement of the value of our work. Below, we respond to each comment in detail.
>
> **Weaknesses:** We include an additional sensitivity analyses in the supplementary PDF Section E.4, where we compared the effects of changing $\tau$ and the ambiguity ball radius $r$ (for RO based methods) on the lenient regret.
>
> **Questions:**
> - (... satisficing threshold ...) For the insulin dosage experiment, we use a safe range of the PBG level as $[K-10, K+10]$ mg/dL (i.e. $-|f(x) - K| \geq -10$), where $K$ represents the PBG target (e.g., set by a clinician). We set $K=110$ which is a rounded approximation of the clinical center of the BG scale $112.5$ mg/dL mentioned in reputable resources in diabetes management [A], [B], that can be regarded as the "safest value". We set $\tau=-10$ for simplicity as it is known that the range [100, 120] mg/dL resides within well-established target PBG ranges stated in clinical guidelines (e.g., [C]). In practice, both $K$ and $\tau$ would be selected by a clinician using domain expertise and clinical assessment of the patient. For the synthetic function, we again set $\tau = -10$, a round value chosen so that behavioral differences of formulations are clearly conveyed. The coincidence has no further significance. We also present an additional experiment in supplementary PDF Section E.1 (parameterized controller selection for an inverted pendulum) that uses $\tau = 0$. In general, $\tau$ should be selected by considering the function maximum and the reward scale, which requires domain expertise in most cases. We highlight that still, in most cases choosing $\tau$ would be more straightforward for the domain expert, in comparison to choosing the ambiguity radius $r$ for the RO approach.
>
>     [A] Kovatchev, Boris P., et al. "Symmetrization of the blood glucose measurement scale and its applications." Diabetes Care 20.11 (1997): 1655-1658.
>
>     [B] Kovatchev, Boris P., et al. "Algorithmic evaluation of metabolic control and risk of severe hypoglycemia in type 1 and type 2 diabetes using self-monitoring blood glucose data." Diabetes Technology \& Therapeutics 5.5 (2003): 817-828.
>
>     [C] American Diabetes Association. "6. Glycemic targets: standards of medical care in diabetes-2021." Diabetes Care 44.Suppl 1 (2021): S73-S84.
>
> - (... selection of the power $p$ ...) The key insight we have of $p$ is that, lower $p$ values give higher importance on the effect of larger perturbations, while higher $p$ values give more importance to the effect of small perturbations. Selection of $p$, thus, depends on the behaviour of the function at the global scale considered together with the nature of the adversary. This suggests that a possible heuristic would be to dynamically adapt $p$ based on the posterior distribution of the GP and distribution of the perturbations of the adversary. However, this may require further assumptions on the nature of the unknown function and behaviour of the adversary.
>
> - (... run time ...) In Section D of the supplementary material we include a discussion about the implementations of the algorithms. In our experiments we worked with discretized domains and the implementation of the acquisition functions of our algorithms have complexity ${\cal O} (N^2)$ where $N = |{\cal X}|$. Notably, this complexity is the same as that of RO-based algorithms.
>
>     To provide an empirical runtime comparison, we ran the insulin dosage experiment again. On our server, while RS-1, RS-2 and RO with $r = 3\cdot\sigma_s$ runs in 173 seconds, RO with $r = \sigma_s$ and RO with $r = \sigma_s / 3$ runs in 172 seconds. This matches our general observation that the runtimes can be considered practically equivalent.

---

> > ### Comment · Reviewer_wsUa · 2025-08-09
> >
> > I thank the authors for detailed response. I will keep the score. I wish the authors the best of luck!

---

### Note · Authors · 2025-08-14

We are sincerely grateful for the time and attention the reviewers and the area chair gave our work. The thoughtful feedback helped us better pose our contributions and improve the presentation. The reviews were overall very positive, recognizing the clarity and merits of our work, and our rebuttals were also met with positive responses.

Reviewer wsUa praised the presentation and noted meaningful experimental improvements over standard RO. Reviewer X7S6 found the paper well-presented with solid experiments; and they mentioned increasing their score to the acceptance side after we addressed their concerns. Reviewer SL7m, considering our work as highly original, commended the balance our RS approach strikes between naive and worst-case methods, our theoretical regret guarantees, and effective numerical comparisons. Reviewer vqo3 appreciated the clarity and well-situated related work, highlighting our novel objectives, solid algorithmic contibution, sublinear regret guarantees.

Below, we recap the discussion with each reviewer:
- **wsUa:** To reviewers' concern on selecting $\tau$, we referred to our sensitivity analysis and argued that choosing a satisficing threshold is more straightforward than selecting an ambiguity radius, and mentioned our insulin dosage experiment to justify. We noted our runtimes and algorithmic complexities are the same of RO methods.
- **X7S6:** Their main concern was on the observation of the perturbed point. We discussed related works they cited and explained that extending our framework to this setting would require assumptions about the adversary’s behavior. We suggested that such an extension would be an interesting future work. Following their question, we presented results for Thompson sampling based RS methods, further showcasing the practical value of our approach.
- **SL7m:** The reviewer had a concern similar to the reviewer X7S6 which we clarified. They also noted the need for additional comparison between our RS approach and other robustness measures. We emphasized that the measures suggested by the reviewer are probabilistic in nature and require modeling perturbations as random variables. We proposed how our methodology could be extended to such a setting, and will include a discussion in the revised paper that addresses the reviewer’s points.
- **vqo3:** Their tone was overall positive. Following the feedback, we decided to motivate our problem setting earlier in the paper using the insulin dosage experiment.

---

### Decision · Program_Chairs · 2025-09-17

**Decision:**

Accept (poster)

**Comment:**

This paper addresses the problem of Gaussian Process optimization under adversarial attacks by adopting robust satisficing as the objective. The authors develop algorithms for two RS formulations based on fragility and critical radius. All reviewers appreciate the clarity and the solid technical contributions of this work.

A concern raised during the rebuttal was that the paper assumes knowledge of the adversarial perturbation. After discussion, the reviewers agreed that this assumption is acceptable and does not diminish the overall contribution. The authors are encouraged to expand the discussion of this limitation in the revision, particularly by comparing their approach with methods in the literature that do not rely on this assumption. In addition, it would be helpful to provide a more detailed discussion of the relationship between the proposed method and other robustness measures.